# MODELING SRP-LSH PERFORMANCE: A THEORETICAL FRAMEWORK FOR OPTIMIZING APPROXIMATE NEAREST NEIGHBOR SEARCH

## ABSTRACT

Approximate nearest neighbor (ANN) search in high-dimensional spaces with sign-random-projection locality-sensitive hashing (SRP-LSH) remains challenging due to the lack of principled approaches for configuring its key parameters. We present a theoretical framework that rigorously models SRP-LSH performance and enables principled parameter configuration. At its core is, to our knowledge, the first analytical model that links the number of hash functions and the Hamming distance threshold to search recall, rooted in the binomial distribution of bit collisions and the angular similarity distribution of vectors. Building upon this model, we develop an adaptive optimization algorithm that minimizes the candidate set size while satisfying user-specified recall targets. Extensive experiments show that our model typically predicts recall with a mean absolute percentage error (MAPE) below 5%. Moreover, our algorithm consistently meets the specified recall targets and simultaneously captures global selectivity trend. Overall, this framework provides a theoretically grounded and practical solution for configuring SRP-LSH in real-world retrieval systems.

## 1 INTRODUCTION

Efficient identification of similar items from large-scale datasets constitutes a core problem across numerous domains, including retrieval-augmented generation (Lewis et al., 2020), data mining (Mo et al., 2022) and machine learning (Hernandez-Jaimes et al., 2024). The primary obstacle to exact nearest neighbor search is the *curse of dimensionality* (Marimont & Shapiro, 1979), whereby increasing dimensionality introduces data sparsity and distance concentration, rendering exact search computationally infeasible.

To address this challenge, approximate nearest neighbor (ANN) search methods (Indyk & Motwani, 1998) have emerged as indispensable tools. These techniques trade a degree of precision for significant gains in efficiency by retrieving points that are within a specified factor $c > 1$ of the true nearest neighbor's distance.

Among the various paradigms for ANN search, *locality-sensitive hashing* (LSH) (Gionis et al., 1999) distinguishes itself as a foundational approach with strong theoretical guarantees. The central idea is to construct hash functions such that nearby points collide with high probability, while distant points collide only rarely. Formally, this intuition is captured by the notion of *sensitivity*. A hash family $\mathcal{H}$ is said to be $(R, cR, p_1, p_2)$-sensitive if for any points $\boldsymbol{x}, \boldsymbol{y}$:

$$\begin{cases} \text{dist}(\boldsymbol{x}, \boldsymbol{y}) \leq R \implies \Pr[h(\boldsymbol{x}) = h(\boldsymbol{y})] \geq p_1, \\ \text{dist}(\boldsymbol{x}, \boldsymbol{y}) > cR \implies \Pr[h(\boldsymbol{x}) = h(\boldsymbol{y})] \leq p_2, \end{cases} \quad (1)$$

where $h \in \mathcal{H}$, $c > 1$, and $p_1 > p_2$. This probabilistic similarity-preserving property has enabled widespread applications of LSH in diverse domains, such as multimedia retrieval (Xu et al., 2021; Xia et al., 2016), anomaly detection (Hamza et al., 2024; Cao et al., 2024), and near-duplicate detection (Gusev & Xu, 2020; Imane et al., 2023).

A prominent and computationally efficient variant of LSH is sign-random-projection LSH (SRP-LSH) (Charikar, 2002). This technique projects high-dimensional vectors onto a set of random

hyperplanes and encodes them into compact binary strings based on the sign of the projections. Similarity is then approximated via the Hamming distance between these binary codes, a process that is highly efficient. Its low storage overhead and computational efficiency have established SRP-LSH a popular choice for industrial-scale systems, particularly in web deduplication (Manku et al., 2007; Kumar et al., 2025).

Despite its widespread adoption, a critical challenge persists: the performance of SRP-LSH is highly sensitive to the selection of its key parameters—the number of hash functions ($m$) and the Hamming distance threshold ($t$) (Manku et al., 2007; Ji et al., 2012). These parameters regulate the trade-off between recall and query efficiency, yet their selection is often a dataset-dependent, empirical process based largely on exhaustive trial-and-error. Crucially, the lack of a theoretical framework—and, in particular, the absence of a quantitative analysis of how these parameters influence retrieval performance—makes it difficult to guarantee parameter configurations that satisfy a specified recall constraint while still improving efficiency.

In this paper, we address this gap by developing a systematic framework for modeling and configuring the parameters of SRP-LSH. Our primary contributions are threefold:

1. We derive an analytical model that explicitly characterizes the relationship between recall and the key SRP-LSH parameters ($m$ and $t$) through collision probability and the angular similarity distribution.

2. We propose an adaptive parameter selection algorithm, grounded in this model, that guarantees the identification of parameter configurations satisfying user-specified recall constraints, while also reducing the candidate set size to enhance query efficiency.

3. We validate the framework on diverse benchmark datasets, achieving recall prediction MAPE below 5%, while consistently meeting specified recall targets and capturing the global selectivity trend, highlighting its practicality for parameter configuration in nearest neighbor search.

## 2 SRP-LSH

We center our analysis on *sign-random-projection LSH* (SRP-LSH) (Charikar, 2002), a binary hashing scheme for angular similarity:

$$\text{sim}(\boldsymbol{a}, \boldsymbol{b}) = 1 - \frac{\theta}{\pi}, \quad \theta = \cos^{-1}\Big(\frac{\langle \boldsymbol{a}, \boldsymbol{b} \rangle}{\|\boldsymbol{a}\|\|\boldsymbol{b}\|}\Big). \tag{2}$$

Given $\mathbf{v} \sim \mathcal{N}(0, I_d)$, the hash function is

$$h_{\mathbf{v}}(\boldsymbol{x}) = \text{sgn}(\mathbf{v}^{\top}\boldsymbol{x}), \quad \text{sgn}(z) = \begin{cases} 1, & z \geq 0 \\ 0, & z < 0. \end{cases} \tag{3}$$

For vectors $\mathbf{a}, \mathbf{b}$, the collision probability equals their similarity:

$$\Pr(h_{\mathbf{v}}(\mathbf{a}) = h_{\mathbf{v}}(\mathbf{b})) = 1 - \frac{\theta_{\mathbf{a},\mathbf{b}}}{\pi}. \tag{4}$$

Concatenating $m$ independent hashes yields $m$-bit codes; Hamming distance serves as the similarity proxy, with a threshold $t$ determining candidate neighbors (Manku et al., 2007; Ji et al., 2012).

While the qualitative effects of $(m, t)$ on recall are illustrated in Figure 1, a quantitative model linking these parameters to recall is missing, making parameter tuning largely empirical. We address this gap by developing a recall model and an adaptive selection algorithm for principled SRP-LSH configuration.

## 3 Modeling Method

Following the analytical framework of (Dong et al., 2008), we model SRP-LSH performance from two perspectives: **recall** and **query cost**. Table 4 summarizes the notation.

**Definition 1** (**Recall**). *The proportion of ground-truth neighbors retrieved:*

$$\rho(\boldsymbol{q}) = \frac{|\mathcal{N}(\boldsymbol{q}) \cap \mathcal{C}(\boldsymbol{q})|}{|\mathcal{N}(\boldsymbol{q})|} \tag{5}$$

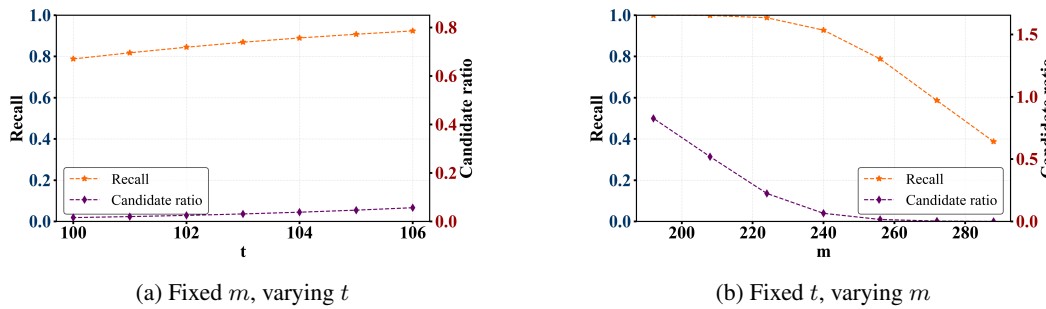

(a) Fixed $m$, varying $t$          (b) Fixed $t$, varying $m$

Figure 1: Effect of $m$ and $t$ on recall–efficiency trade-off.

**Definition 2 (Selectivity).** *A proxy for query cost, defined as the candidate ratio:*

$$\tau(\boldsymbol{q}) = \frac{|\mathcal{C}(\boldsymbol{q})|}{|\mathcal{D}|} \tag{6}$$

In SRP-LSH, query cost is mainly determined by scanning retrieved candidates, so selectivity—the candidate fraction—serves as a platform-agnostic proxy for efficiency (Dong et al., 2008).

We measure average performance as expectations:

$$\rho = \mathbb{E}_{\boldsymbol{q}\sim\mathcal{Q}}[\rho(\boldsymbol{q})], \quad \tau = \mathbb{E}_{\boldsymbol{q}\sim\mathcal{Q}}[\tau(\boldsymbol{q})] \tag{7}$$

where $\mathcal{Q}$ is the query set.

Since retrieval probability depends on the angle $\theta = \mathrm{Angular}(\boldsymbol{q},\boldsymbol{p})$, we define the *angular recall function*:

$$\rho(\theta) = \Pr[\boldsymbol{p} \in \mathcal{C}(\boldsymbol{q}) \mid \mathrm{Angular}(\boldsymbol{q},\boldsymbol{p}) = \theta] \tag{8}$$

Selectivity and recall can both be expressed as expectations over angle distributions:

$$\tau = \mathbb{E}_{\theta\sim g(\theta)}[\rho(\theta)], \quad \rho = \frac{1}{n}\sum_{k=1}^{n}\mathbb{E}_{\theta_k\sim g_k(\theta_k)}[\rho(\theta_k)], \tag{9}$$

where $g(\theta)$ is the random angle distribution and $g_k(\theta_k)$ is the $k$-th neighbor angle distribution.

Thus, modeling $\rho$ and $\tau$ requires:

1. The analytical form of $\rho(\theta)$;
2. The distributions of random angles $\theta$ and nearest-neighbor angles $\theta_k$.

### 3.1 MODELING THE RECALL FUNCTION

Given a Hamming distance threshold $t$, a data vector $\boldsymbol{p}$ is considered recalled if $\mathrm{Hamming}(h_{1,\ldots,m}(\boldsymbol{q}), h_{1,\ldots,m}(\boldsymbol{p})) \leq t$. Conditioned on the angular distance $\theta = \mathrm{Angular}(\boldsymbol{q},\boldsymbol{p})$, the recall probability is

$$\rho(\theta) = \Pr[\mathrm{Hamming}(h_{1,\ldots,m}(\boldsymbol{q}), h_{1,\ldots,m}(\boldsymbol{p})) \leq t \mid \mathrm{Angular}(\boldsymbol{q},\boldsymbol{p}) = \theta]. \tag{10}$$

**Theorem 1 (Hamming Distance in SRP-LSH is Binomially Distributed).** *Let $\theta$ be the angle between $\boldsymbol{q}$ and $\boldsymbol{p}$. Each hash bit disagrees with probability $\frac{\theta}{\pi}$, and agrees with probability $1 - \frac{\theta}{\pi}$. With $m$ independent hash functions, the Hamming distance $X$ follows*

$$X \sim \mathrm{Binomial}\left(m, \frac{\theta}{\pi}\right) \tag{11}$$

The proof of Theorem 1 is provided in Appendix A.2.

Hence, $\rho(\theta)$ equals the cumulative distribution function (CDF) of this binomial distribution at $t$:

$$\rho(\theta) = \Pr[X \leq t] = \sum_{i=0}^{t}\binom{m}{i}\left(\frac{\theta}{\pi}\right)^{i}\left(1 - \frac{\theta}{\pi}\right)^{m-i} \tag{12}$$

## 3.2 MODELING THE DISTRIBUTION OF ANGLES

To extend the recall model to real datasets, we characterize two angular quantities: (i) $\theta$, the angle between random vectors, and (ii) $\theta_k$, the angle between a query and its $k$-th nearest neighbor.

**Theorem 2** (**Normalized Cosine Similarity Between Random Unit Vectors is Beta-Distributed**). *Let $\mathbf{a}, \mathbf{b}$ be independent samples from the unit hypersphere in $\mathbb{R}^d$ ($d \geq 2$), and let $\theta \in (0, \pi)$ denote their angle. Define the normalized cosine similarity $Y = \frac{\cos\theta + 1}{2}$. Then*

$$Y \sim \text{Beta}(\alpha, \beta),$$

*where $(\alpha, \beta)$ are the shape parameters of the Beta distribution, estimated by maximum likelihood estimation (MLE).*

The proof of Theorem 2 is provided in Appendix A.3.

By a change of variables, the probability density function (PDF) of the random angle $\theta$ is

$$g(\theta) = f_Y\left(\tfrac{\cos\theta+1}{2}\right) \cdot \left|\tfrac{d}{d\theta}\left(\tfrac{\cos\theta+1}{2}\right)\right| = f_Y\left(\tfrac{\cos\theta+1}{2}\right) \cdot \tfrac{\sin\theta}{2}, \quad \theta \in (0, \pi), \tag{13}$$

where $Y \sim \text{Beta}(\alpha, \beta)$ and $f_Y(\cdot)$ denotes its PDF.

Following (Dong et al., 2008), the normalized cosine similarity of the $k$-th nearest neighbor is represented as

$$Y_k = \tfrac{\cos\theta_k+1}{2} \sim \text{Beta}(\alpha_k, \beta_k).$$

where $\theta_k$ denotes the angle between the query and the $k$-th nearest neighbor, and $(\alpha_k, \beta_k)$ are the Beta distribution parameters, which can be estimated via maximum likelihood estimation.

The corresponding probability density function (PDF) of the angle $\theta_k$ is

$$g_k(\theta_k) = f_{Y_k}\left(\tfrac{\cos\theta_k+1}{2}\right) \cdot \tfrac{\sin\theta_k}{2}, \quad \theta_k \in (0, \pi), \tag{14}$$

where $f_{Y_k}(\cdot)$ denotes the Beta PDF of $Y_k$.

A detailed discussion covering both arbitrary vector pairs and the $k$-th neighbor case is deferred to Appendix A.7.

The expected recall and selectivity then become

$$\rho = \tfrac{1}{n}\sum_{k=1}^{n}\int_0^\pi \rho(\theta_k)\, g_k(\theta_k)\, d\theta_k, \quad \tau = \int_0^\pi \rho(\theta)\, g(\theta)\, d\theta. \tag{15}$$

Substituting $\rho(\theta)$ from Eq. 12 gives explicit forms for $\rho$ and $\tau$:

$$\rho = \tfrac{1}{n}\sum_{k=1}^{n}\int_0^\pi \sum_{i=0}^{t}\binom{m}{i}\left(\tfrac{\theta_k}{\pi}\right)^i\left(1 - \tfrac{\theta_k}{\pi}\right)^{m-i} f_{Y_k}\left(\tfrac{\cos\theta_k+1}{2}\right)\tfrac{\sin\theta_k}{2}\, d\theta_k, \tag{16}$$

$$\tau = \int_0^\pi \sum_{i=0}^{t}\binom{m}{i}\left(\tfrac{\theta}{\pi}\right)^i\left(1 - \tfrac{\theta}{\pi}\right)^{m-i} f_Y\left(\tfrac{\cos\theta+1}{2}\right)\tfrac{\sin\theta}{2}\, d\theta. \tag{17}$$

For large datasets, exact $g_k(\theta_k)$ is intractable. Following (Slaney et al., 2012), we estimate it via subsampling:

**Lemma 1** (**Subsampling the Retrieval Dataset Yields a Lower Bound on the CDF of Nearest Neighbor Angle Distributions**). *Let $g_k^l(\theta_k)$ denote the nearest-neighbor angle distribution estimated from a subsampled retrieval dataset, and $g_k(\theta_k)$ the corresponding distribution derived from the full retrieval dataset. Then, for all $\theta_k \in [0, \pi]$, the following inequality holds:*

$$\int_0^{\theta_k} g_k^l(\phi)\, d\phi \leq \int_0^{\theta_k} g_k(\phi)\, d\phi.$$

The proof of Lemma 1 is provided in Appendix A.4.

This guarantees pessimistic recall estimates: subsampling avoids optimistic bias and ensures recall targets, at the cost of slightly higher query cost.

---

**Algorithm 1** FINDOPTIMALPARAMETERS: Search for optimal parameter combination

---

**Require:** Search range for the number of hash functions: $m \in [m_{\text{low}}, m_{\text{high}}]$, step size $step$, target recall $recall$, number of true nearest neighbors $n$, retrieval dataset $\mathcal{D}$, query set $\mathcal{Q}$

**Ensure:** Optimal parameter pair $(m^*, t^*)$

1: $Ans \leftarrow [\,]$
2: **for** $m^* \leftarrow m_{\text{low}}$ **to** $m_{\text{high}}$ **step** $step$ **do**
3:      $t_{\text{left}} \leftarrow 1, \quad t_{\text{right}} \leftarrow m^*, \quad t^* \leftarrow 0$
4:      **while** $t_{\text{left}} \leq t_{\text{right}}$ **do**
5:          $t_{\text{mid}} \leftarrow \lfloor \frac{t_{\text{left}} + t_{\text{right}}}{2} \rfloor$
6:          $r \leftarrow$ CALCULATERECALL 2$(\mathcal{Q}, \mathcal{D}, m^*, t_{\text{mid}}, n)$
7:          **if** $r \geq recall$ **then**
8:             $t^* \leftarrow t_{\text{mid}}$
9:             $t_{\text{right}} \leftarrow t_{\text{mid}} - 1$
10:         **else**
11:            $t_{\text{left}} \leftarrow t_{\text{mid}} + 1$
12:         **end if**
13:      **end while**
14:      $r \leftarrow$ CALCULATERECALL$(\mathcal{Q}, \mathcal{D}, m^*, t^*, n)$
15:      $s \leftarrow$ CALCULATESELECTIVITY 3$(\mathcal{D}, m^*, t^*)$
16:      Append $(m^*, t^*, r, s)$ to $Ans$
17: **end for**
18: Sort $Ans$ in ascending order of selectivity
19: $(m^*, t^*, r, s) \leftarrow Ans[0]$
20: **return** $(m^*, t^*)$

---

## 4    OPTIMAL PARAMETER SELECTION METHOD

Subsampling the retrieval set shifts the estimated distribution rightward, yielding conservative recall. In contrast, query subsampling preserves unbiased estimates of the mean and variance of angular distances:

**Lemma 2 (Subsampling the Query set Yields Unbiased Estimation of Nearest Neighbor Angular Statistics).** *For query set $\mathcal{Q}$, the sample mean and variance from any subset $\mathcal{S} \subseteq \mathcal{Q}$ are unbiased estimators of the population values:*

$$\mathbb{E}[\bar{\theta}] = \mu, \mathbb{E}[\hat{\sigma}^2] = \sigma^2$$

*where $\hat{\sigma}^2 = \frac{1}{n-1} \sum_{i=1}^{n} (\theta_i - \bar{\theta})^2$, and $n$ is the sample size.*

The proof of Lemma 2 is provided in Appendix A.5.

Thus, we estimate the angular distance distribution using a small query subset. The optimal parameters minimize selectivity under a recall constraint:

$$
\begin{aligned}
\min_{m,\, t} &\quad \text{Selectivity}(m, t) \\
\text{s.t.} &\quad \text{Recall}_{\text{pred}}(m, t) \geq \text{Target Recall}.
\end{aligned}
\tag{18}
$$

Algorithm 1 efficiently solves this by iterating over $m$ and performing binary search over $t$, selecting the pair with smallest selectivity.

## 5    EXPERIMENTS

### 5.1    PREDICTION EXPERIMENTAL SETUP

We validate the effectiveness of the proposed framework on six benchmark datasets (see Appendix A.7, Table 5), focusing on recall prediction and parameter selection for SRP-LSH. All experiments are conducted on a server running Ubuntu 22.04.5, equipped with Spark 3.3.1, Python 3.10, and Rust 1.87.0, on an AMD EPYC 7R32 CPU (16 GB JVM memory, 4 threads).

The ground-truth top-$k$ recall is computed using our SRP-LSH implementation for $k \in \{10, 50, 100\}$, with results reported as the mean and standard deviation across 1,000 queries. For each dataset, the entire database is used as the retrieval dataset, and 1,000 vectors are randomly sampled as the query set.

Recall prediction is performed by modeling the angular distribution between each query and its $k$-th true nearest neighbor, obtained via exhaustive search. Selectivity is estimated by sampling 20,000 vector pairs from the retrieval dataset and modeling their angular distances.

To assess prediction precision, we vary the number of hash functions $m$ and the Hamming distance threshold $t$ by fixing one parameter while sweeping the other. To evaluate robustness with respect to query and retrieval dataset sizes, we conduct four experimental configurations: (i) **Exp. 1**, 10k queries (1k for `gist-960`) with the full retrieval dataset; (ii) **Exp. 2**, 100 queries with the full retrieval dataset; (iii) **Exp. 3**, 100 queries with 50% of the retrieval dataset; and (iv) **Exp. 4**, 100 queries with 10% of the retrieval dataset. For `deep-image-96`, the retrieval dataset is further reduced to 5% and 1% in Exp. 3 and Exp. 4, respectively, to control computational cost.

Prediction accuracy is measured using the *Mean Absolute Error* (MAE) and the *Mean Absolute Percentage Error* (MAPE), defined as

$$\text{MAE} = \frac{1}{n} \sum_{i=1}^{n} |y_i - \hat{y}_i|, \tag{19}$$

$$\text{MAPE} = \frac{100\%}{n} \sum_{i=1}^{n} \left| \frac{y_i - \hat{y}_i}{y_i} \right|, \tag{20}$$

where $y_i$ denotes the ground-truth value in the $i$-th experiment, $\hat{y}_i$ denotes the corresponding prediction, and $n$ is the total number of experiments. These metrics jointly capture both absolute and relative prediction errors, enabling a fair evaluation across datasets with varying scales and distributions.

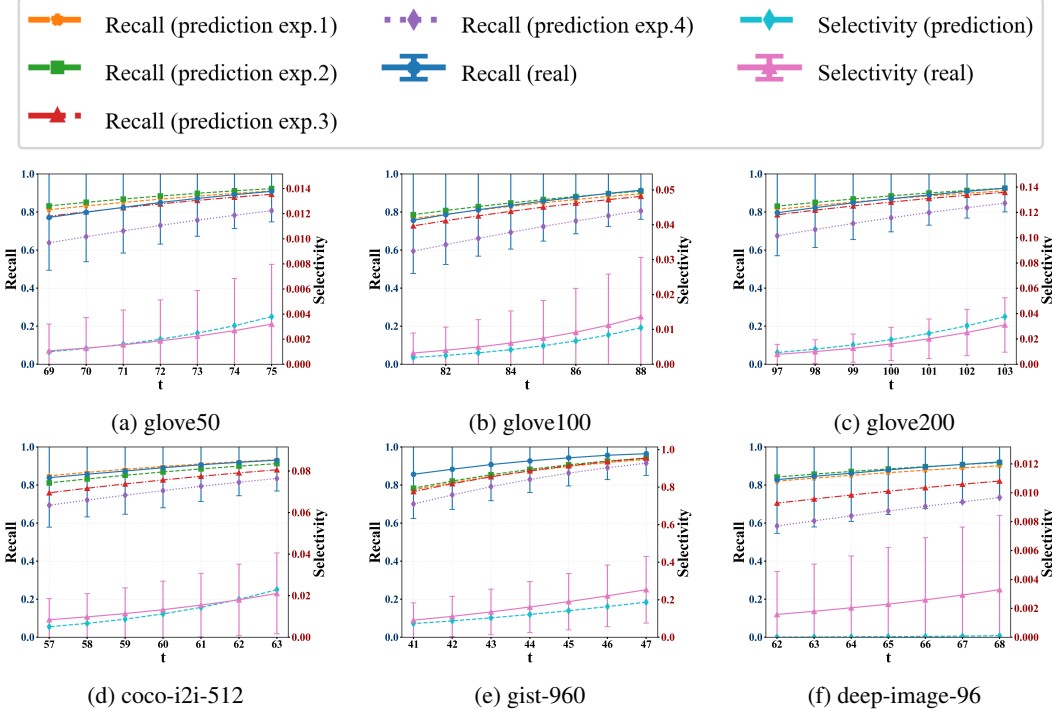

Figure 2: Predicted and empirical recall and selectivity under fixed $m = 256$ and varying $t$ across six datasets. Error bars indicate one standard deviation.

Table 1: Prediction accuracy of recall and selectivity across datasets under different Top-$k$ settings.

| Dataset | topk | Recall | | Selectivity | |
|---|---|---|---|---|---|
| | | MAPE(%) | MAE | MAPE(%) | MAE |
| **glove-50-angular** | 10 | 2.415% | 0.01960 | 8.976% | 0.00021 |
| | 50 | 2.181% | 0.01760 | 20.282% | 0.00091 |
| | 100 | 2.054% | 0.01670 | 25.304% | 0.00154 |
| **glove-100-angular** | 10 | 1.066% | 0.00920 | 30.700% | 0.00212 |
| | 50 | 1.644% | 0.01440 | 21.960% | 0.00367 |
| | 100 | 1.818% | 0.01580 | 19.132% | 0.00462 |
| **glove-200-angular** | 10 | 1.024% | 0.00876 | 20.315% | 0.00363 |
| | 50 | 0.954% | 0.00808 | 20.600% | 0.00680 |
| | 100 | 0.908% | 0.00747 | 20.265% | 0.00813 |
| **coco-i2i-512** | 10 | 0.738% | 0.00645 | 18.187% | 0.00204 |
| | 50 | 0.436% | 0.00359 | 13.083% | 0.00252 |
| | 100 | 0.485% | 0.00394 | 13.448% | 0.00341 |
| **gist-960** | 10 | 5.983% | 0.05426 | 23.899% | 0.04082 |
| | 50 | 5.275% | 0.04720 | 24.810% | 0.04899 |
| | 100 | 4.799% | 0.03748 | 25.375% | 0.05745 |
| **deep-image-96** | 10 | 1.595% | 0.01413 | 97.973% | 0.00231 |
| | 50 | 1.113% | 0.00985 | 96.387% | 0.00324 |
| | 100 | 1.300% | 0.01152 | 95.135% | 0.00404 |

## 5.2 ANALYSIS OF THE THEORETICAL MODEL

We first assess the predictive performance of our model under the setting of $m = 256$ hash functions and Top-$k = 10$, varying the Hamming distance threshold $t$. For each dataset, we identify the threshold achieving target recall $0.8$ and compare predicted versus empirical recall and selectivity in its vicinity. Results on six benchmarks are shown in Figure 2.

As shown in Figure 2, predicted recall closely tracks empirical measurements across all datasets, confirming the accuracy and generalizability of our model. Subsampling queries produces nearly identical curves, validating the unbiasedness of angular statistics (Lemma 2) and demonstrating that a small query subset suffices to approximate the full distribution at much lower cost. Four observations follow:

- **High accuracy (Exp. 1 & Exp. 2).** The predictions closely match empirical recall on full datasets, indicating the model reliably captures nearest-neighbor behavior.

- **Lower-bound property (Exp. 3 & Exp. 4).** On reduced datasets, recall is slightly underestimated but stays within one standard deviation.

- **Practical efficiency.** Accurate predictions can be obtained from as few as 100 queries, greatly reducing the cost of angle estimation for large datasets.

- **Selectivity trends.** Exact selectivity is harder to predict when values are extremely small, but the model captures global trends and deviations remain within statistical error (Dong et al., 2008).

We further quantify the accuracy of our predictions across different Top-$k$ values, we report the results of Experiment 1 in Table 1. Recall predictions are highly accurate, with mean absolute percentage error (MAPE) below $5\%$ on most datasets. The largest deviation occurs on gist-960, likely due to its high dimensionality, while large-scale datasets such as glove-200 and deep-image-96 yield errors close to $1\%$, highlighting scalability. Selectivity predictions exhibit higher relative error because absolute values are very small (often $10^{-3}$–$10^{-4}$), but the absolute errors (MAE) remain within practical bounds.

In summary, our model provides reliable recall–selectivity predictions for SRP-LSH, enabling efficient and robust parameter selection (see Appendix A.8 for additional results).

Table 2: Comparison of different distribution fitting methods across datasets. Lower values indicate better approximation.

| Dataset | Distribution | MAPE (Selectivity) | MAPE (Recall) |
|---|---|---|---|
| **glove-50** | Gamma | 152.162% | 5.193% |
| | Chi-square | 152.162% | 5.098% |
| | Empirical | 42.399% | 5.820% |
| | **Beta(ours)** | **25.304%** | **4.449%** |
| **glove-100** | Gamma | 2.411% | 0.982% |
| | Chi-square | **2.409%** | 0.823% |
| | Empirical | 19.024% | 0.959% |
| | **Beta(ours)** | 19.132% | **0.510%** |
| **glove-200** | Gamma | 24.925% | 3.152% |
| | Chi-square | 24.925% | 3.152% |
| | Empirical | 21.624% | 2.957% |
| | **Beta(ours)** | **20.265%** | **2.280%** |
| **coco-i2i-512** | Gamma | 20.039% | 4.289% |
| | Chi-square | 20.039% | 4.694% |
| | Empirical | 34.695% | 4.743% |
| | **Beta(ours)** | **13.448%** | **4.027%** |
| **gist-960** | Gamma | **17.140%** | 4.295% |
| | Chi-square | **17.140%** | 21.344% |
| | Empirical | 20.028% | 4.408% |
| | **Beta(ours)** | 25.375% | **4.171%** |
| **deep-image-96** | Gamma | 83.552% | 1.973% |
| | Chi-square | 83.552% | 1.963% |
| | Empirical | **19.635%** | 1.573% |
| | **Beta(ours)** | 95.153% | **0.843%** |

## 5.3 COMPARISON WITH ALTERNATIVE DISTRIBUTION FITTING

To evaluate the validity of our proposed Beta-based distribution model for SRP-LSH, we conduct a comparative study with alternative distribution fitting approaches. In particular, we consider methods inspired by prior work on Multi-probe LSH (Dong et al., 2008; Slaney et al., 2012), which were originally developed to model the distribution of Euclidean distances between vectors. We adapt these methods to approximate the distribution of angular distances (i.e., cosine similarity) in SRP-LSH, allowing a direct comparison with our Beta-based model. Additionally, we include empirical probability densities computed directly from observed cosine similarity counts.

For a fair comparison, all competing methods are applied in the cosine similarity setting. We set Top-$k = 100$ and compute the mean absolute percentage error (MAPE) for both recall and selectivity, under the same query and dataset configuration as Experiment 2. Results are summarized in Table 2.

The Beta model consistently achieves **the lowest recall errors** on all evaluated datasets, reducing MAPE by 6–46% relative to the strongest baselines, demonstrating robust modeling of angular similarity distributions. For selectivity, Beta remains competitive or superior in most scenarios. Although the gamma distribution attains slightly lower selectivity error on `gist-960`, it fails to capture recall accurately; similarly, while empirical distributions better approximate selectivity on `deep-image-96`, Beta still provides the most accurate recall predictions. Overall, these results demonstrate that the Beta distribution offers the most reliable and generalizable approximation of cosine similarity in SRP-LSH.

## 5.4 ANALYSIS OF THE PARAMETER SELECTION METHOD

We further demonstrate the practical utility of our model by proposing a binary search-based algorithm (Algorithm 1) to automatically identify SRP-LSH configurations that minimize selectivity

Table 3: Comparison of fixed settings and adaptive configurations

| Dataset | topk | recall (prediction) | recall (real) | parameter search time(s) |
|---|---|---|---|---|
| **glove-50-angular** | 10 | 0.9051 | 0.9015 | 0.8832 |
| | 50 | 0.9069 | 0.9092 | 3.0536 |
| | 100 | 0.9017 | 0.9157 | 5.6941 |
| **glove-100-angular** | 10 | 0.9140 | 0.9201 | 1.1211 |
| | 50 | 0.9104 | 0.9254 | 3.1222 |
| | 100 | 0.9107 | 0.9281 | 5.7506 |
| **glove-200-angular** | 10 | 0.9025 | 0.9166 | 2.2701 |
| | 50 | 0.9039 | 0.9208 | 3.0829 |
| | 100 | 0.9030 | 0.9159 | 5.9016 |
| **coco-i2i-512** | 10 | 0.9088 | 0.8904 | 2.0128 |
| | 50 | 0.9089 | 0.8951 | 3.7692 |
| | 100 | 0.9047 | 0.9050 | 6.3883 |
| **gist-960** | 10 | 0.9045 | 0.9208 | 7.1361 |
| | 50 | 0.9032 | 0.9116 | 9.0401 |
| | 100 | 0.9062 | 0.9123 | 11.0612 |
| **deep-image-96** | 10 | 0.9016 | 0.9245 | 11.7792 |
| | 50 | 0.9000 | 0.9210 | 14.2691 |
| | 100 | 0.9042 | 0.9220 | 16.0076 |

while satisfying a target recall. Given a search range for $m$ and a target recall, the algorithm iteratively selects the $(m, t)$ pair with the lowest selectivity that still achieves the recall constraint.

To evaluate the algorithm, we design two groups of experiments. In the first group, we set $m_{\text{low}} = 128$, $m_{\text{high}} = 320$, and $step = 16$ to search for parameter configurations that satisfy recall $\geq 0.9$ for $k = 10, 50, 100$, and the results are summarized in Table 3. In the second group, we analyze how the selectivity varies with $m$ within a small local interval to better understand the selectivity trend, and the result is shown in Figure 3.

We first evaluate the algorithm across multiple datasets and $k$ values. Table 3 reports the true recall and the predicted recall under the parameter configurations selected by our algorithm, along with the corresponding parameter search time. The reported search time is based on a naive implementation of our method and can be further reduced with standard engineering optimizations. Across all datasets and $k$ values, the adaptive algorithm attains recall $\geq 0.9$ in the majority of cases, with only minor deviations (e.g., $k = 10, 50$ on `coco-i2i-512`, where recall is approximately 0.89). Overall, the results demonstrate that the algorithm provides accurate recall predictions and efficiently identifies effective parameter configurations.

To illustrate that our algorithm can identify parameter settings with minimal selectivity, Figure 3 presents how selectivity varies with $m$ within a local interval. As shown in the figure, selectivity does not decrease monotonically as $m$ increases. This behavior is expected, because our goal is to find parameter configurations that achieve a recall value greater than or equal to a specified target, and the recall is jointly determined by both $m$ and $t$ rather than being a continuous function of $m$ alone. In our experiments, we further observe cases where, even under the same recall level, a smaller $m$ yields lower selectivity than a larger $m$, highlighting the importance of careful parameter selection. The figure shows that the selectivity trend predicted by our algorithm closely aligns with the true selectivity trend, and the algorithm accurately identifies the configuration that achieves the minimal selectivity. These results demonstrate that our method effectively captures how selectivity varies with the parameters, providing reliable guidance for parameter configuration.

## 5.5 SUMMARY OF EXPERIMENTAL FINDINGS

Our experiments lead to three key takeaways:

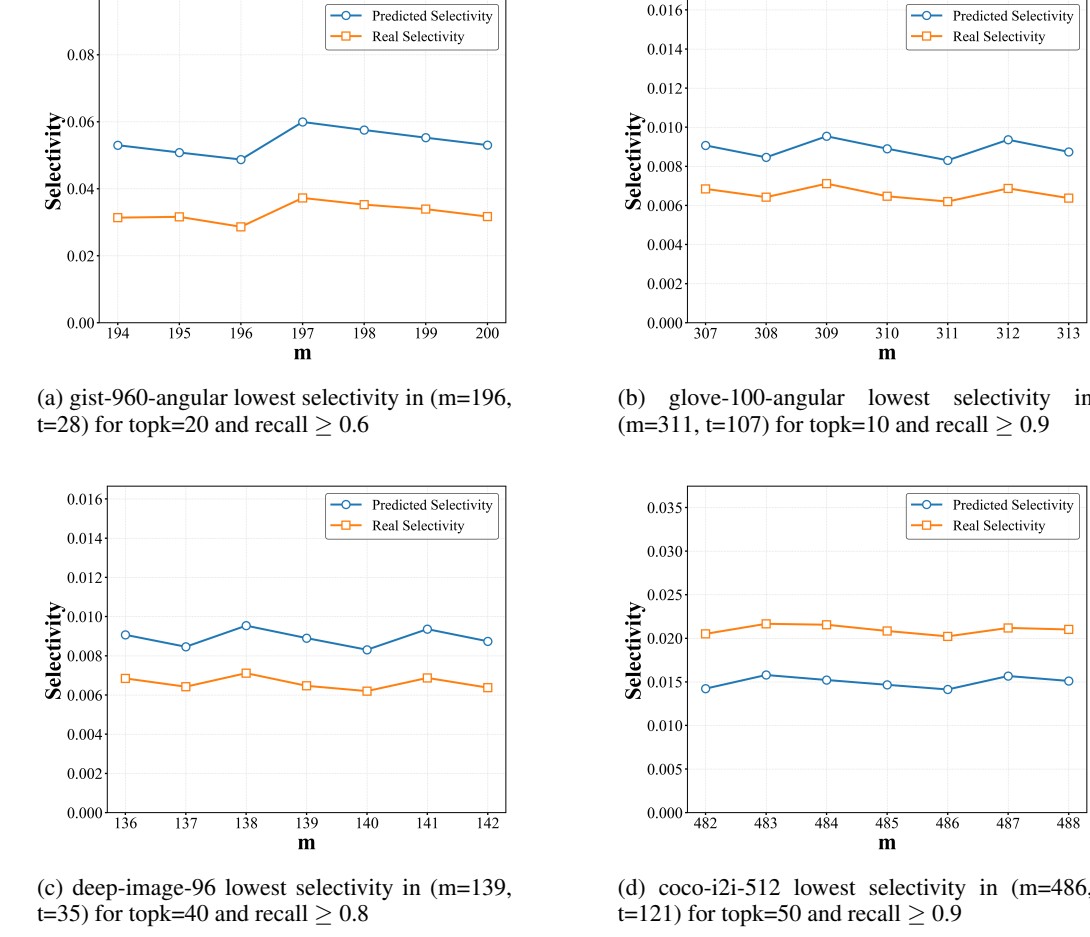

(a) gist-960-angular lowest selectivity in (m=196, t=28) for topk=20 and recall $\geq 0.6$

(b) glove-100-angular lowest selectivity in (m=311, t=107) for topk=10 and recall $\geq 0.9$

(c) deep-image-96 lowest selectivity in (m=139, t=35) for topk=40 and recall $\geq 0.8$

(d) coco-i2i-512 lowest selectivity in (m=486, t=121) for topk=50 and recall $\geq 0.9$

Figure 3: Selectivity Trends Across datasets Under Recall Constraints

- The proposed model accurately predicts recall and selectivity across datasets and top-$k$ settings, with recall highly reliable (MAPE $< 5\%$) and selectivity trends well captured.

- Modeling cosine similarity with a Beta distribution outperforms gamma, chi-square, and empirical alternatives, providing a more robust and generalizable fit.

- The adaptive parameter selection algorithm exploits the model to automatically identify efficient SRP-LSH configurations, reducing selectivity without loss of recall.

Together, these results demonstrate both the theoretical soundness and the practical utility of the framework for efficient, accurate large-scale vector retrieval.

## 6 CONCLUSION

In this work, we propose a principled framework for SRP-LSH, comprising an analytical modeling method that characterizes recall–parameter relationships and an efficient optimization algorithm for scalable parameter tuning in high-dimensional vector retrieval systems. Extensive experiments show that the framework generalizes effectively across diverse datasets and retrieval scenarios. Its combination of predictive accuracy, robustness, and low computational overhead makes it well-suited for practical deployment in large-scale similarity search applications.

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

# A APPENDIX

## A.1 RELATED WORK

**Parameter Optimization in LSH.** Parameter optimization remains a longstanding fundamental challenge in the LSH literature. Prior work has investigated both analytical and empirical approaches, including statistical performance models and parameter derivations for Multi-probe LSH (Lv et al., 2007; Dong et al., 2008; Slaney et al., 2012) which rely on bucket-probing strategies and empirical distance distributions. Additionally, dynamic bucketing strategies have been proposed to enhance hash table efficiency (Tian et al., 2023). However, these techniques are inherently tailored to their respective mechanisms and cannot be directly applied to SRP-LSH, which is based on random hyperplane projections and bit collisions. As a result, a systematic, analytically rigorous framework for modeling the parameter-performance trade-off in SRP-LSH is still lacking, thereby hindering principled parameter selection and often necessitating heuristic or trial-and-error approaches.

**Learning-Driven Adaptive ANN Search.** Concurrent research has explored leveraging machine learning to enhance approximate nearest neighbor (ANN) search. These approaches span from learning improved hash functions and representations (Deng et al., 2024; Tan et al., 2020; Huo et al., 2024) to dynamically adapting search strategies at query time via techniques such as early termination (Li et al., 2020; 2025) or intelligent partition selection (Zeng et al., 2025; Mohoney et al., 2025). While powerful, these methods often suffer from high training costs and dependencies on specific index architectures. In contrast, we focus on a lightweight and theoretically grounded parameter optimization model for the classic SRP-LSH.

**Hybrid Methods and Extended ANN Scenarios.** The scope of ANN has expanded to address more complex retrieval tasks, including those with attribute constraints (Wang et al., 2023), range filters (Liang et al., 2024), or specialized data types such as high-dimensional trajectories (Deng et al., 2024). Hybrid systems have emerged that integrate LSH with graph-based indices (Zhao et al., 2023), locality-sensitive filtering (Pham & Liu, 2022), or near-memory processing (Li et al., 2025) to enhance performance. These advanced frameworks typically target specialized use cases or introduce architectural dependencies that diverge from the core LSH mechanism.

**Positioning Our Contribution.** We develop a theoretical model that explicitly characterizes the relationship between recall and the key parameters $(m, t)$ in SRP-LSH, thereby elucidating the inherent trade-off between recall and computational cost. This formulation enables a principled algorithm for automated parameter configuration, achieving high efficiency without incurring the training overhead or architectural constraints of complex learning-based systems.

## A.2 PROOF OF THEOREM 1

*Proof.* Let $\theta \in [0, \pi]$ denote the angle between the query vector $\boldsymbol{q}$ and a data vector $\boldsymbol{p}$. In SRP-LSH, each hash function corresponds to an independently and randomly sampled hyperplane through the origin. As established in Theorem 1, the probability that a single hash bit differs between $h(\boldsymbol{q})$ and $h(\boldsymbol{p})$ equals $\frac{\theta}{\pi}$, with the agreement probability being $1 - \frac{\theta}{\pi}$ (Charikar, 2002).

Given the independence of the $m$ hash functions, each bit-wise comparison can be modeled as an independent Bernoulli trial with success probability $\frac{\theta}{\pi}$ (interpreted as a mismatch event). Let $X$ represent the Hamming distance between the $m$-bit hash codes $h_{1,\ldots,m}(\boldsymbol{q})$ and $h_{1,\ldots,m}(\boldsymbol{p})$. Since $X$ counts the number of mismatches across $m$ independent trials, we have:

$$X \sim \text{Binomial}\left(m, \frac{\theta}{\pi}\right),$$

which directly verifies the assertion of Theorem 1. $\square$

**Empirical Validation** We empirically evaluate the validity of Eq. 12 by generating 1,000 vector pairs in $\mathbb{R}^{256}$ with target pairwise angles drawn uniformly from the interval $\theta \in (0, \pi)$. For each sampled pair, 1,000 Monte Carlo trials are conducted to estimate the corresponding recall. Figure 4 compares the theoretical predictions from Eq. 12 with simulation results under two representative parameter configurations.

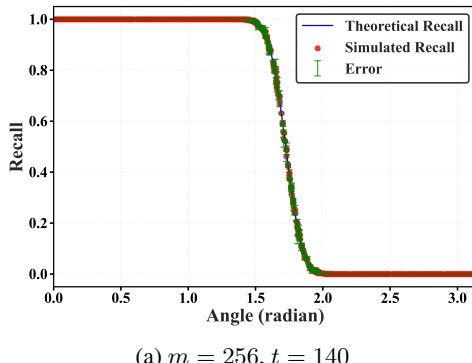 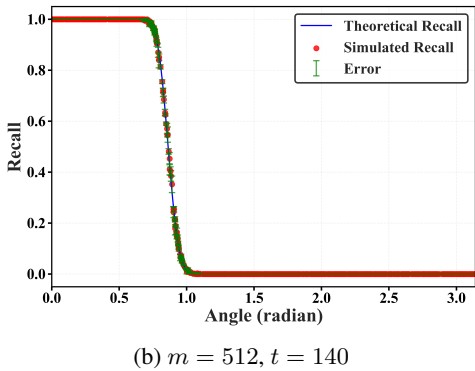

(a) $m = 256$, $t = 140$           (b) $m = 512$, $t = 140$

Figure 4: Comparison between theoretical and empirical recall under different parameter settings.

**Discussion** As illustrated in Figure 4, the theoretical recall derived from Eq. 12 exhibits close alignment with empirical estimates across varying parameter settings. This consistency confirms that Eq. 12 accurately characterizes the relationship among recall, angle $\theta$, the number of hash functions m, and the Hamming distance threshold t. The results provide robust empirical support for the analytical formulation and validate its applicability to practical retrieval scenarios.

### A.3 PROOF OF THEOREM 2

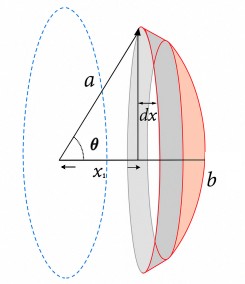

Figure 5: Geometric illustration of the angle $\theta$ between two vectors on the unit sphere

*Proof.* Building upon the analysis of cosine similarity distributions in (Kia, 2016), consider two vectors $\mathbf{a}$ and $\mathbf{b}$ independently and uniformly sampled from the $(d-1)$-dimensional unit hypersphere, denoted as $\mathbb{S}^{d-1}$ (i.e., the set of points in $\mathbb{R}^d$ at unit distance from the origin), as illustrated in Figure 5. Let $\theta$ denote the angle between them.

Without loss of generality, we fix $\mathbf{b}$ along the $x$-axis, i.e.,

$$\mathbf{b} = (1, 0, \cdots, 0), \quad \mathbf{a} = (x_1, x_2, \cdots, x_d).$$

Since both $\mathbf{a}$ and $\mathbf{b}$ are unit vectors, it follows that

$$\cos \theta = \mathbf{a} \cdot \mathbf{b} = x_1.$$

Hence, $x_1$ (hereafter denoted by $x$) corresponds to the $x$-coordinate of $\mathbf{a}$. Geometrically, the locus of points on the unit hypersphere forming the same angle $\theta$ with $\mathbf{b}$ is a $(d-2)$-dimensional hypersphere orthogonal to the $x$-axis, with radius $\sqrt{1 - x^2}$. Therefore, the probability density of $x$ is proportional to the surface area of this hypersphere.

Let $S_{d-1}(1)$ denote the surface area of a unit $(d-1)$-dimensional hypersphere. Then the surface area of a $(d-1)$-dimensional hypersphere of radius $r$ is

$$S_{d-1}(r) = S_{d-1}(1) \, r^{d-1}.$$

Accordingly, for a $(d-2)$-dimensional hypersphere of radius $\sqrt{1-x^2}$, the surface area is

$$S_{d-2}(\sqrt{1-x^2}) = S_{d-2}(1)\,(1-x^2)^{\frac{d-2}{2}}.$$

Consider an infinitesimal band around $x = \cos\theta$ of thickness $dx$. The corresponding surface area element on the $(d-1)$-dimensional unit sphere is

$$dA(x) = S_{d-2}(\sqrt{1-x^2})\frac{dx}{\sqrt{1-x^2}} = S_{d-2}(1)\,(1-x^2)^{\frac{d-3}{2}}\,dx.$$

The probability density function of the transformed variable

$$Y = \frac{\cos\theta + 1}{2} \in (0,1), \quad x = 2y - 1.$$

is obtained by differentiating the cumulative probability:

$$\begin{aligned}
f_Y(y) &= \frac{d}{dy}\Pr[Y \leq y] \\
&= \frac{d}{dy}\int_0^y \frac{dA(x)}{S_{d-1}(1)} \quad \text{with } x = 2t-1 \\
&= \frac{d}{dy}\int_0^y \frac{2\,S_{d-2}(1)(1-(2t-1)^2)^{\frac{d-3}{2}}}{S_{d-1}(1)}dt \\
&= 2 \cdot \frac{S_{d-2}(1)}{S_{d-1}(1)}(1-(2y-1)^2)^{\frac{d-3}{2}} \\
&= 2 \cdot \frac{S_{d-2}(1)}{S_{d-1}(1)}(4y(1-y))^{\frac{d-3}{2}} \\
&= 2^{d-2} \cdot \frac{S_{d-2}(1)}{S_{d-1}(1)}y^{\frac{d-3}{2}}(1-y)^{\frac{d-3}{2}} \\
&= 2^{d-2} \cdot \frac{S_{d-2}(1)}{S_{d-1}(1)}\underbrace{y^{\frac{d-1}{2}-1}(1-y)^{\frac{d-1}{2}-1}}
\end{aligned}$$

From the above derivation, it is evident that $f_Y(y)$ corresponds to the probability density function of a Beta distribution for $y \in (0,1)$.

The case of two independent vectors uniformly sampled from the hypersphere can be viewed as a special instance of a more general situation, where a fixed vector is paired with an arbitrary vector on the hypersphere. In this general setting, we assume that the normalized cosine similarity also follows a Beta distribution with parameters $(\alpha, \beta)$, i.e.,

$$Y = \frac{\cos\theta + 1}{2} \sim \text{Beta}(\alpha, \beta).$$

Although this is an approximation, it is commonly adopted in high-dimensional analysis, and the resulting error is negligible in practice (Kia, 2016; Dong et al., 2008).

Since $Y = \frac{\cos\theta+1}{2}$ is strictly monotonic and differentiable for $\theta \in (0, \pi)$, the change-of-variable formula yields the probability density function of $\theta$:

$$g(\theta) = f_Y\left(\frac{\cos\theta + 1}{2}\right) \cdot \left|\frac{d}{d\theta}\left(\frac{\cos\theta + 1}{2}\right)\right| = f_Y\left(\frac{\cos\theta + 1}{2}\right) \cdot \frac{\sin\theta}{2}, \quad \theta \in (0,\pi), \quad (21)$$

where $f_Y(y)$ denotes the probability density function of $\text{Beta}(\alpha, \beta)$:

$$f_Y(y; \alpha, \beta) = \frac{y^{\alpha-1}(1-y)^{\beta-1}}{B(\alpha, \beta)}, \quad y \in (0,1), \quad (22)$$

with $B(\alpha, \beta)$ representing the Beta function ensuring normalization.

This derivation establishes the analytical form of the angular distribution between two independent vectors uniformly distributed on the $d$-dimensional unit hypersphere. By analogy, we approximate the angle $\theta_k$ between a given query vector and its $k$-th nearest neighbor as following a similar distribution, enabling estimation of nearest-neighbor angular statistics. □

Table 4: Notation and definitions

| Symbol | Description |
| --- | --- |
| $\mathcal{D}$ | Retrieval dataset (i.e., the retrieval database) |
| $\mathcal{Q}$ | Query set |
| $\mathcal{N}(\boldsymbol{q})$ | Ground-truth set of top-$n$ nearest neighbors of query $\boldsymbol{q}$ |
| $\mathcal{C}(\boldsymbol{q})$ | Candidate set returned for query $\boldsymbol{q}$ (includes ground-truth neighbors) |
| $\theta_k$ | Angle (in radians) between query vector and its $k$-th nearest neighbor |
| $\theta$ | Angle (in radians) between query vector and any data vector |
| $m$ | Number of hash functions |
| $t$ | Hamming distance threshold |
| $\rho(\boldsymbol{q})$ | Recall of query $\boldsymbol{q}$ |
| $\tau(\boldsymbol{q})$ | Selectivity of query $\boldsymbol{q}$ |
| $\mathrm{Angular}(\boldsymbol{q}, \boldsymbol{p})$ | Angular distance (in radians) between vectors $\boldsymbol{q}$ and $\boldsymbol{p}$ |
| $n$ | Number of nearest neighbors |
| $\mathrm{Hamming}(\cdot)$ | Hamming distance between binary hash codes |
| $g(\theta)$ | Angular distribution between query vector and any data vector |
| $g_k(\theta_k)$ | Angular distribution between query vector and its $k$-th nearest neighbor |

We employ the maximum likelihood estimation (MLE) to infer the parameters $\alpha$ and $\beta$ of the Beta distribution. Given observed samples $\{x_i\}_{i=1}^n$, the likelihood function is expressed as:

$$L(\alpha, \beta) = \prod_{i=1}^n \frac{x_i^{\alpha-1}(1 - x_i)^{\beta-1}}{B(\alpha, \beta)}, \tag{23}$$

where $B(\alpha, \beta)$ denotes the Beta function.

Taking the logarithm, the log-likelihood function simplifies to

$$\ln L(\alpha, \beta) = (\alpha - 1) \sum_{i=1}^n \ln x_i + (\beta - 1) \sum_{i=1}^n \ln(1 - x_i) - n \ln B(\alpha, \beta). \tag{24}$$

Setting the partial derivatives of $\ell(\alpha, \beta)$ with respect to $\alpha$ and $\beta$ to zero yields the following system of equations:

$$\begin{cases} S_1 = n \left[ \psi(\alpha) - \psi(\alpha + \beta) \right], \\ S_2 = n \left[ \psi(\beta) - \psi(\alpha + \beta) \right], \end{cases} \tag{25}$$

where

$$S_1 = \sum_{i=1}^n \ln x_i, \quad S_2 = \sum_{i=1}^n \ln(1 - x_i), \tag{26}$$

and $\psi(\cdot)$ denotes the digamma function, defined as the logarithmic derivative of the Gamma function:

$$\psi(z) = \frac{d}{dz} \ln \Gamma(z) = \frac{\Gamma'(z)}{\Gamma(z)}. \tag{27}$$

Since the system in Eq. equation 25 lacks a closed-form solution, we resort to numerical optimization techniques—such as Newton's method—to estimate $\alpha$ and $\beta$.

### A.4 PROOF OF LEMMA 1

*Proof.* Let $\mathcal{D}$ denote the full dataset, and let $\mathcal{S} \subseteq \mathcal{D}$ be a subset obtained via random sampling. For a fixed query vector $\boldsymbol{q} \in \mathbb{R}^d$, define the nearest neighbor angles in the full dataset and the subset as

$$\theta_{\mathrm{true}} = \min_{\boldsymbol{p} \in \mathcal{D}} \mathrm{Angular}(\boldsymbol{q}, \boldsymbol{p}), \qquad \theta_{\mathrm{sub}} = \min_{\boldsymbol{p} \in \mathcal{S}} \mathrm{Angular}(\boldsymbol{q}, \boldsymbol{p}).$$

Since $\mathcal{S} \subseteq \mathcal{D}$, it follows immediately that

$$\theta_{\mathrm{true}} \leq \theta_{\mathrm{sub}}.$$

Let $F_{\text{true}}$ and $F_{\text{sub}}$ denote the cumulative distribution functions (CDFs) of $\theta_{\text{true}}$ and $\theta_{\text{sub}}$, respectively, i.e.,

$$F_{\text{true}}(x) = \Pr[\theta_{\text{true}} \leq x], \qquad F_{\text{sub}}(x) = \Pr[\theta_{\text{sub}} \leq x].$$

By the monotonicity of probability measures under set inclusion, for any $x \in [0, \pi]$ we have

$$F_{\text{sub}}(x) = \Pr[\theta_{\text{sub}} \leq x]$$

$$= \Pr\left[\min_{\boldsymbol{p} \in \mathcal{S}} \text{Angular}(\boldsymbol{q}, \boldsymbol{p}) \leq x\right]$$

$$\leq \Pr\left[\min_{\boldsymbol{p} \in \mathcal{D}} \text{Angular}(\boldsymbol{q}, \boldsymbol{p}) \leq x\right]$$

$$= F_{\text{true}}(x)$$

Finally, by definition of the integrals of $g_k^l(\theta_k)$ and $g_k(\theta_k)$ as functions of the CDFs $F_{\text{sub}}$ and $F_{\text{true}}$, it follows that for any interval $[0, \theta_k]$,

$$\int_0^{\theta_k} g_k^l(\phi) \, d\phi \leq \int_0^{\theta_k} g_k(\phi) \, d\phi,$$

which establishes the claimed lower bound. $\qquad\qquad\square$

To empirically validate the theoretical result of lemma 1, we conducted an experiment on the Glove-200 dataset.

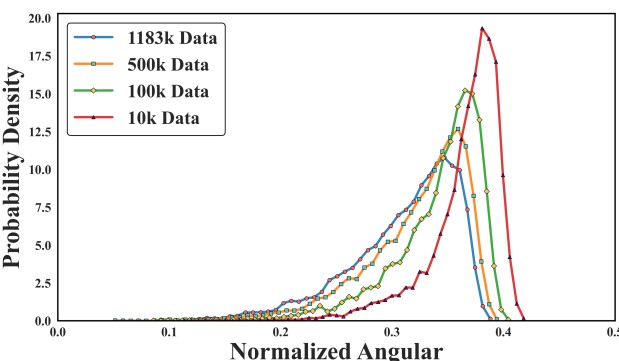

Figure 6: Distributions of the top-1 nearest neighbor *normalized angular distance* $\frac{\theta}{\pi}$ for subsets of varying sizes in the `GloVe-200` dataset. As the subsampling ratio decreases, the distribution shifts rightward, indicating a statistically larger $\theta$ between the retrieved nearest neighbors and the query vectors. This trend aligns with the theoretical prediction in Lemma 1.

Figure 6 clearly demonstrates this phenomenon: as the subset size decreases, the empirical distribution of nearest neighbor angles $\theta$ shifts toward larger values, accompanied by an increase in the mean angle. This observation confirms that nearest neighbors obtained from subsampled datasets constitute a statistical lower bound relative to those from the full dataset.

### A.5 PROOF OF LEMMA2

*Proof.* Let the full query set be denoted by $\mathcal{T}$, with cardinality $n_t$. Consider a subset $\mathcal{F} \subset \mathcal{T}$ of size $n_f$, randomly sampled from $\mathcal{T}$. Define the random variable $\theta$ as the nearest neighbor angular distance computed over queries in the full set $\mathcal{T}$, with population mean $\mu$ and variance $\sigma^2$. The distances obtained from the subset $\mathcal{F}$, denoted $\theta_1, \theta_2, \cdots, \theta_{n_f}$, are treated as i.i.d. samples drawn from the distribution of $\theta$.

For the sample mean $\bar{\theta} = \frac{1}{n_f} \sum_{i=1}^{n_f} \theta_i$, linearity of expectation immediately gives

$$\mathbb{E}[\bar{\theta}] = \frac{1}{n_f} \sum_{i=1}^{n_f} \mathbb{E}[\theta_i] = \mu,$$

showing that $\bar{\theta}$ is unbiased.

For the sample variance

$$S^2 = \frac{1}{n_f - 1} \sum_{i=1}^{n_f} (\theta_i - \bar{\theta})^2,$$

expanding the quadratic term yields

$$\sum_{i=1}^{n_f} (\theta_i - \bar{\theta})^2 = \sum_{i=1}^{n_f} (\theta_i - \mu)^2 - n_f(\bar{\theta} - \mu)^2.$$

Taking expectations gives

$$\mathbb{E}\left[\sum_{i=1}^{n_f} (\theta_i - \bar{\theta})^2\right] = n_f\sigma^2 - n_f\mathrm{Var}(\bar{\theta}) = (n_f - 1)\sigma^2,$$

hence

$$\mathbb{E}[S^2] = \sigma^2.$$

Thus both the sample mean and the sample variance are unbiased estimators of their population counterparts. □

In summary, regardless of the size of the subset $n_f$, both the sample mean and the sample variance computed from the subset $\mathcal{F}$ are unbiased estimators of their population counterparts in expectation. This result justifies that modeling based on query subsets does not introduce systematic estimation bias.

### A.6 ADDITIONAL ALGORITHMS FOR PARAMETER SELECTION

---

**Algorithm 2** CALCULATERECALL: Compute Recall

---

**Require:** Number of hash functions $m$, Hamming distance threshold $t$, number of true nearest neighbors $n$, retrieval dataset $\mathcal{D}$, query set $\mathcal{Q}$
**Ensure:** Recall $recall$
1: Compute recall according to Eq. 16: $recall \leftarrow \rho(\mathcal{Q}, \mathcal{D}, m, t, n)$
2: **return** $recall$

---

**Algorithm 3** CALCULATESELECTIVITY: Compute Selectivity

---

**Require:** Number of hash functions $m$, Hamming distance threshold $t$, retrieval dataset $\mathcal{D}$
**Ensure:** Selectivity $selectivity$
1: Compute selectivity according to Eq. 17: $selectivity \leftarrow \tau(\mathcal{D}, m, t)$
2: **return** $selectivity$

---

To complement the main algorithm, we provide two auxiliary procedures that directly implement the analytical performance formulas introduced in the main text. These procedures offer a lightweight alternative for evaluating SRP-LSH configurations without relying on empirical trials.

Algorithm 2 estimates the expected recall via Eq. 16, given the retrieval dataset $\mathcal{D}$, query set $\mathcal{Q}$, the number of hash functions $m$, the Hamming distance threshold $t$, and the number of true nearest neighbors $n$. Algorithm 3 computes the expected selectivity—i.e., the proportion of dataset points retrieved—via Eq. 17 with $m$, $\mathcal{D}$ and $t$ as inputs. Together, these algorithms enable efficient and reproducible evaluation of SRP-LSH parameter settings.

### A.7 FURTHER ANALYSIS OF THE DISTRIBUTION OF NORMALIZED COSINE SIMILARITY BETWEEN VECTOR PAIRS

To further validate the proposed modeling framework, we conduct a statistical analysis of the angular relationships between randomly sampled vector pairs drawn from multiple benchmark datasets.

Table 5: Statistics of experimental datasets.

| Dataset | Database Size | Dimension | Queries |
|---|---|---|---|
| glove-50 | 1.18M | 50 | 10k |
| glove-100 | 1.18M | 100 | 10k |
| glove-200 | 1.18M | 200 | 10k |
| coco-i2i-512 | 113k | 512 | 10k |
| deep-image-96 | 9.99M | 96 | 10k |
| gist-960 | 1.00M | 960 | 1k |

Data source: ANN-Benchmarks (Aumüller et al., 2020).

Specifically, for each dataset, we randomly sample 100,000 vector pairs and compute their normalized cosine similarity defined as

$$Y = \frac{\cos\theta + 1}{2}.$$

where $\theta$ denotes the angle between the vectors. The empirical probability density of $y$ is estimated using histograms, and a Beta distribution is fitted via maximum likelihood estimation (MLE):

$$Y \sim \text{Beta}(\alpha, \beta).$$

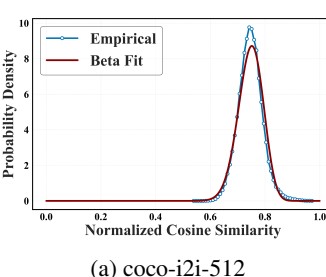 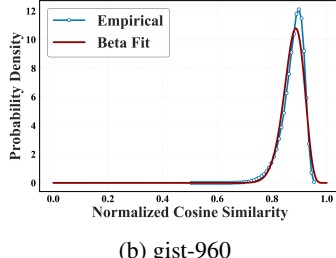 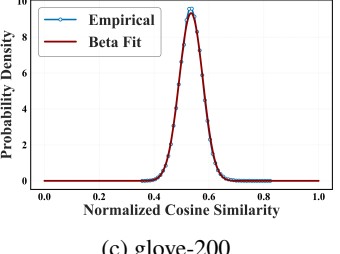

(a) coco-i2i-512        (b) gist-960        (c) glove-200

Figure 7: Comparison of the distributions of normalized cosine similarity between arbitrary vector pairs

The motivation for this modeling choice is twofold. First, the distribution of cosine similarity directly impacts the probability of retrieving nearest neighbors under SRP-LSH. Second, adopting a flexible parametric model like the Beta distribution allows us to capture both the concentration and spread of similarity values while preserving analytical tractability for subsequent theoretical derivations. As shown in Figure 7, although empirical distributions may deviate from the uniform hypersphere assumption due to dataset-specific structures, the Beta distribution provides a close approximation to the observed trends. This justifies its use as a surrogate model for the random variable $Y = \frac{\cos\theta + 1}{2}$.

We further investigate the angular distribution characteristics between vectors and their nearest neighbors. Let $Y_k = \frac{\cos\theta_k + 1}{2}$, where $\theta_k$ denotes the angle between a test vector and its $k$-th nearest neighbor. Experiments are conducted on a query set containing $10,000$ query vectors, disjoint from the retrieval dataset. For each query vector, we retrieve its Top-1, Top-10, Top-50, and Top-100 nearest neighbors and compute the corresponding values of $Y_k$.

Figure 8 presents histograms of $Y_k$ for different neighbor ranks $k$, overlaid with fitted Beta distributions. The Beta distribution effectively captures the empirical trends, and the similarity distribution systematically shifts toward lower values as $k$ increases. Importantly, substantial variation across datasets indicates that no single distribution universally can characterize the similarity of nearest neighbors. Therefore, accurately modeling $\theta_k$ necessitates dataset-specific Beta fitting, ensuring both precise theoretical analysis and practical applicability in similarity search.

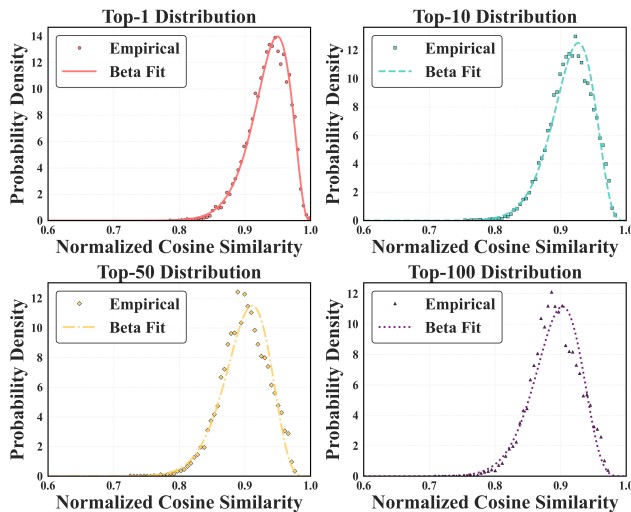

Figure 8: Probability density distribution of the normalized cosine similarity $y$ between query vectors and their Top-$k$ nearest neighbors

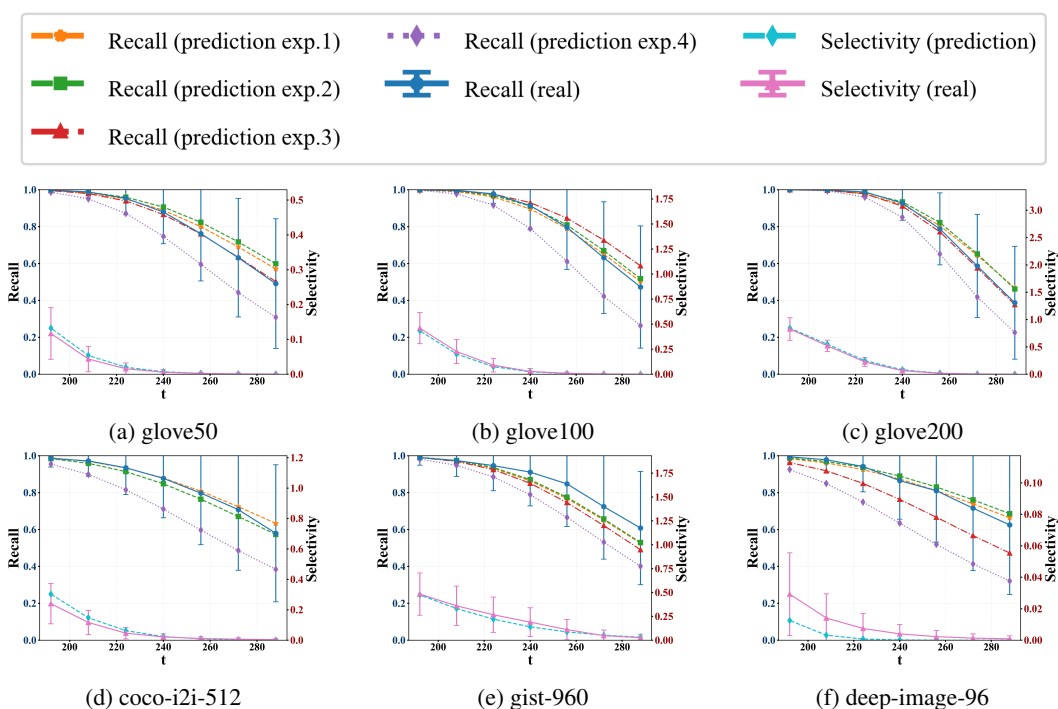

Figure 9: Variation of recall and selectivity with respect to the number of hash functions $m$, under a fixed Hamming distance threshold $t$. Error bars indicate one standard deviation.

## A.8 ADDITIONAL EXPERIMENTS ON PREDICTION MODEL

To further assess the robustness and predictive accuracy of our framework, we conduct an additional set of experiments focusing on three aspects: (i) the influence of the number of hash functions $m$ on predictive performance and (ii) the effect of varying the retrieval dataset coverage when estimating nearest-neighbor similarity distributions. (iii) the impact of query set variation, evaluated by repeating the experiments on multiple randomly sampled query subsets to confirm robustness against query selection. (iv) the generalization capability of our framework under substantially larger val-

Table 6: Prediction accuracy across different retrieval dataset coverage levels.

| Dataset | topk | Query Scope (100%) | | Query Scope (50%) | | Query Scope (10%) | |
|---|---|---|---|---|---|---|---|
| | | MAPE(%) | MAE | MAPE(%) | MAE | MAPE(%) | MAE |
| **glove-50-angular** | 10 | 4.373% | 0.03599 | 0.933% | 0.00810 | 14.149% | 0.11870 |
| | 50 | 4.559% | 0.03740 | 1.349% | 0.01160 | 17.817% | 0.14840 |
| | 100 | 4.449% | 0.03660 | 1.181% | 0.01530 | 19.572% | 0.16300 |
| **glove-100-angular** | 10 | 1.544% | 0.01240 | 3.752% | 0.03150 | 16.472% | 0.13710 |
| | 50 | 0.614% | 0.00512 | 4.752% | 0.04080 | 17.993% | 0.15410 |
| | 100 | 0.510% | 0.00437 | 5.014% | 0.04340 | 18.693% | 0.16160 |
| **glove-200-angular** | 10 | 1.933% | 0.01613 | 1.927% | 0.01682 | 11.688% | 0.10024 |
| | 50 | 2.092% | 0.01748 | 2.273% | 0.01962 | 13.178% | 0.11251 |
| | 100 | 2.280% | 0.01890 | 2.424% | 0.02060 | 14.410% | 0.11980 |
| **coco-i2i-512** | 10 | 2.529% | 0.02230 | 7.311% | 0.06450 | 13.627% | 0.12030 |
| | 50 | 3.632% | 0.03110 | 9.338% | 0.07970 | 20.624% | 0.17596 |
| | 100 | 4.027% | 0.03380 | 10.594% | 0.08870 | 24.909% | 0.20860 |
| **gist-960** | 10 | 5.090% | 0.04606 | 6.037% | 0.05453 | 11.053% | 0.10010 |
| | 50 | 4.581% | 0.04091 | 6.757% | 0.06047 | 13.468% | 0.12103 |
| | 100 | 4.171% | 0.03748 | 6.433% | 0.05797 | 13.340% | 0.12070 |
| **deep-image-96** | 10 | 0.750% | 0.00641 | 12.779% | 0.11680 | 24.669% | 0.21550 |
| | 50 | 1.017% | 0.00852 | 15.172% | 0.13060 | 30.854% | 0.26580 |
| | 100 | 0.843% | 0.00714 | 16.525% | 0.14330 | 33.865% | 0.29370 |

ues of $m$, verifying whether its predictive behavior remains stable and accurate beyond the range evaluated in the main experiments.

**Influence of the Number of Hash Functions.** We first investigate the effect of varying $m$ while fixing the Hamming distance threshold $t$. The results in Figure 9 show that recall decreases monotonically as $m$ increases when Top-$k = 50$. This behavior is expected: increasing $m$ lengthens the hash code, thereby tightening the collision probability and reducing the likelihood that true neighbors falling within the same Hamming radius.

Across all datasets and parameter settings, predicted recall values lie within one standard deviation of the empirical measurements. Moreover, the predicted and empirical curves exhibit strong agreement, particularly in the high-recall regime, which is critical for recall-sensitive applications where overestimation could harm system reliability. Selectivity predictions also align closely with empirical trends and remain within one standard deviation of the empirical measurements, with particularly high accuracy in the low-recall region, indicating the model's utility in guiding accuracy–efficiency trade-offs.

**Effect of Retrieval Dataset Coverage.** Table 6 reports prediction errors when modeling nearest-neighbor similarity distributions using varying proportions of the retrieval dataset, with evaluation conducted on a fixed query set of 100 queries. For `deep-image-96`, we consider retrieval coverage levels of 100%, 5%, and 1%, while for the remaining datasets we adopt 100%, 50%, and 10%. Prediction accuracy consistently improves as retrieval coverage increases, since larger retrieval datasets more faithfully approximate the underlying nearest-neighbor angle distribution.

Notably, estimates derived from smaller retrieval subsets act as *pessimistic* predictions—serving as lower bounds on the true recall—which can be beneficial in real-world scenarios where recall variability is high. Furthermore, subsampling drastically reduces the computational cost of estimating neighbor angles, which would otherwise require exhaustive nearest-neighbor search. This renders conservative estimation via partial coverage both computationally efficient and practically robust, especially in practical deployments.

**Impact of Query Set Variation** To eliminate potential concerns that different random seeds or possible overlaps between the query set used for modeling and the query set used for measuring empirical recall might affect the results, we conduct three additional experiments: (i) queries are

Table 7: MAPE of predicted recall under different query set variations: (i) seed=42, (ii) seed=1235, and (iii) disjoint queries.

| Dataset | Top-$k$ | Exp1 (42) | Exp2 (1234) | Exp3 (disjoint) |
|---|---|---|---|---|
| **glove-50-angular** | 10 | 4.373% | 1.277% | 3.694% |
| | 50 | 4.559% | 0.874% | 3.097% |
| | 100 | 4.449% | 0.687% | 3.015% |
| **glove-100-angular** | 10 | 1.544% | 2.573% | 1.354% |
| | 50 | 0.614% | 3.278% | 1.890% |
| | 100 | 0.510% | 3.391% | 1.988% |
| **glove-200-angular** | 10 | 1.933% | 1.026% | 2.430% |
| | 50 | 2.092% | 0.919% | 1.861% |
| | 100 | 2.280% | 0.864% | 1.423% |
| **coco-i2i-512** | 10 | 2.529% | 3.898% | 2.298% |
| | 50 | 3.632% | 5.744% | 2.130% |
| | 100 | 4.027% | 3.430% | 2.153% |
| **deep-image-96** | 10 | 0.750% | 1.529% | 1.502% |
| | 50 | 1.017% | 1.503% | 1.025% |
| | 100 | 0.843% | 1.929% | 1.241% |
| **gist-960** | 10 | 5.090% | 5.308% | 3.864% |
| | 50 | 4.581% | 3.512% | 2.778% |
| | 100 | 4.171% | 2.958% | 2.145% |

sampled with random seed 42, (ii) queries are sampled with random seed 1234, and (iii) the modeling queries and the validation queries are enforced to be completely disjoint. The results are reported in the Table 7, and the consistency across these settings further confirms the robustness of our framework.

The results in Table 7 indicate that different query set configurations (random seeds or disjoint sampling) have only a minor impact on predictive accuracy. The consistently low MAPE across datasets and settings highlights the robustness of our theoretical model and confirms that its accuracy is not sensitive to the choice of query set.

**Generalization under Large $m$**  To further assess the robustness of our framework beyond the parameter regimes used in the main experiments, we evaluate its generalization ability under substantially larger numbers of hash functions $m$. For four representative datasets, we consider values of $m$ ranging from 512 up to 2056—significantly exceeding those in the primary evaluation—and report the corresponding recall prediction errors in terms of MAE and MAPE (Table 8).

Across all datasets and top-$k$ configurations, the prediction performance remains consistently strong. The MAE values remain below 0.06 in all cases, and the MAPE values largely fall within the range of 1–7%, exhibiting no monotonic increase or systematic degradation as $m$ grows. Even at the upper end of the tested range (e.g., $m > 2000$), the estimator achieves accuracy comparable to that observed in the standard settings.

These findings indicate that our framework as a whole is not sensitive to the specific choice of hash budget. Instead, it generalizes robustly to substantially larger values of $m$, providing strong empirical evidence that it captures an intrinsic and stable relationship between hash collisions and recall, independent of the absolute hashing dimensionality.

A.9  LLM USAGE

After completing the initial draft, we employed ChatGPT as a writing assistant to refine the manuscript. Specifically, we used it to polish the Introduction and Related Work sections with the goal of improving clarity, coherence, and academic expression. The tool was used solely for language refinement, such as enhancing readability, improving the flow of arguments, and ensur-

Table 8: Evaluation of the generalization ability of our framework when applied to substantially larger values of $m$. Results are reported on multiple datasets and top-$k$ configurations using recall MAE and MAPE as metrics.

| Dataset | Top-$k$ | m | recall(MAE) | recall(MAPE) |
|---|---|---|---|---|
| **glove-200-angular** | 10 | 512 | 0.008 | 1.006% |
| | 30 | 786 | 0.039 | 6.102% |
| | 50 | 1024 | 0.045 | 3.409% |
| **coco-i2i-512** | 10 | 1024 | 0.056 | 6.064% |
| | 20 | 2056 | 0.013 | 1.851% |
| | 40 | 1512 | 0.061 | 7.145% |
| **glove-100-angular** | 50 | 1024 | 0.007 | 0.891% |
| | 20 | 1600 | 0.059 | 6.174% |
| | 100 | 1256 | 0.033 | 4.979% |
| **glove-50-angular** | 20 | 2000 | 0.014 | 1.665% |
| | 30 | 1400 | 0.046 | 4.759% |
| | 40 | 1600 | 0.026 | 3.066% |

ing consistency in terminology and style, without generating new scientific content or altering the substantive contributions of the work.

