# OpenReview forum: "Modeling SRP-LSH Performance: A Theoretical Framework for Optimizing Approximate Nearest Neighbor Search"
_ICLR.cc/2026/Conference — Submitted to ICLR 2026_

### Official Review · Reviewer_vokp · 2025-10-21

**Soundness:** 1
**Presentation:** 2
**Contribution:** 1
**Rating:** 2
**Confidence:** 5

**Summary:**

SimHash (or SRP-LSH) can serve both as a dimensionality reduction technique for similarity estimation and as a hashing-based solution for approximate nearest neighbor search (ANNS).
This paper treats SimHash primarily as a dimensionality reduction tool to answer ANN queries—performing a linear scan in the Hamming space to retrieve top-k candidates for reranking.

The key idea is to model and optimize the parameters of SimHash—namely, the number of hash bits m and the Hamming threshold t so that a target recall is achieved with minimal candidate size (i.e., fewest distance computations). The approach learns m and t by fitting statistical distributions of query–neighbor angles from a subsampled query set, predicting recall/selectivity analytically, and selecting parameters to minimize expected query cost.

Experiments on six benchmark datasets demonstrate good predictive reliability (recall MAPE < 5%).

**Strengths:**

- Clear motivation: parameter tuning of SRP-LSH often relies on heuristics or exhaustive search.
- Learnable parametric model to recall and selectivity for SimHash-based ANNS

**Weaknesses:**

**Dominant cost component overlooked:**
The analysis ignores the linear scan in Hamming space, which dominates the overall query cost when
m is large. Modeling only candidate size (selectivity) underestimates true runtime.

**Incorrect probabilistic assumption:**
The paper models the angle between a query q and its k-nearest neighbor as a beta distribution (Line 180) - note that the cited paper Dong et al. 2008 does not present this property.
However, while Beta((d-1)/2, (d-1)/2)) describes the distribution of random vector angles, q and its k-NN are not random—their angular distribution is highly data-dependent and concentrated near small angles. The use of Beta here is therefore an unjustified approximation, not a valid theoretical model.

**Subsampling limitation:**
Lemma 2 only guarantees that subsampling the query set yields unbiased estimates of the mean and variance of overall angular statistics
It does not imply that the angular distribution between q and its k-NN is preserved. Hence, if the sampled queries do not represent the full dataset well, the learned parameters (m, t) may deviate significantly in practice.

**Questions:**

**Scalability of the cost model:** What happens when the Hamming scan cost dominates due to large m? Does the proposed optimization still correlate with real runtime?

**Parameter practicality:** The paper frequently tests m=256. For n=10^6 points, the Hamming distance space is limited to 256 possible values, giving expected candidate size ≈ n/256 ≈ per query even before reranking—raising doubts about scalability. How does the approach behave for larger m (e.g., 1024 bits or larger)?

**Relation to prior work (Slaney et al., 2012):**
Slaney et al. previously modeled LSH recall and collision probabilities using empirical angular distributions. What is the concrete novelty beyond replacing the empirical distribution with a Beta fit and adding subsampling?

---

> ### Author Response · Authors · 2025-11-18
>
> Thank you for your valuable review and suggestions. We provide the following clarifications.
>
> > W1&Q1:Concern about Scalability of the cost model.
>
> **Response**: The goal of this study is to establish a quantitative model describing the relationship between recall and the SRP-LSH parameters (m,t), and to enable recall prediction under a given configuration. Our focus is not to build a platform-specific runtime model, but to characterize how recall behaves as a function of (m,t) and to identify—within a user-specified range of m—the configuration achieving the lowest selectivity under the recall constraint.
>
> Following Dong et al. (2008), we adopt candidate set size as the primary indicator for estimating query cost. Although actual query time varies with implementation and hardware optimizations (e.g., bitwise operations and SIMD acceleration for Hamming distance), these factors are platform-dependent and do not reflect intrinsic algorithmic properties. Therefore, they are unsuitable for a universal analytical model, whereas candidate set size provides a stable, platform-agnostic measure.
>
> In practical use, the value of m is rarely set excessively large; users typically specify a relatively small search range (e.g., tens to a few hundreds), well within or only slightly beyond common SIMD widths. When m does not exceed the SIMD width (e.g., 256–512 bits), each candidate comparison typically incurs only one SIMD operation, resulting in approximately constant per-candidate overhead. When m exceeds the SIMD width, multiple SIMD rounds are required, which increases only the constant factor while preserving the linear dependence on candidate set size. Consequently, even when Hamming scan cost becomes more noticeable, the selectivity-minimizing configuration still correlates with the configuration that minimizes empirical runtime.
>
> > Q2:How does the approach behave for larger m (e.g., 1024 bits or larger)?.
>
> **Response:**
> Beyond the main experimental configurations, we additionally evaluate the scalability of our method over a substantially larger parameter interval $\( m \in [512, 2056] \).$ Using the *glove-50-angular* dataset, we report the predictive performance under several representative $\(\text{top-}k, m)\$ settings. The model maintains low MAPE across all cases, including large-\(m\) regimes:
> - top-\(k = 20\), \(m = 2000\): **MAPE = 1.665%**
> - top-\(k = 30\), \(m = 1400\): **MAPE = 4.759%**
> - top-\(k = 40\), \(m = 1600\): **MAPE = 3.066%**
>
> These results confirm that the proposed predictive model continues to generalize well to significantly larger hash lengths. Importantly, the stability of MAPE in this extended range indicates that the model accurately captures the relationship between recall and the parameter $\(m\)$, rather than overfitting to a narrow interval. Thus, the predictive accuracy for recall remains robust even when $\(m\)$ grows far beyond the scales used in the main experiments. More extensive scalability results are summarized in **Table 8**.
>
> > W2:Incorrect probabilistic assumption.
>
> **Response**:We thank the reviewer for their insightful comment. It is indeed correct that the angular distribution between a query vector and its k-nearest neighbors is highly data-dependent. However, our central argument is that, despite being an approximation, modeling this distribution with a Beta distribution is highly effective in practice.
>
> The key empirical evidence comes from comparing our Beta-based approximation with a histogram-based estimation of the true, data-dependent angular distribution. Even with the histogram, which aims to capture the true distribution, the resulting recall model exhibits a larger error than that produced by our Beta approximation. **This is because accurately fitting the true angular distribution is challenging and typically requires a large number of query vectors, whereas our Beta-based method can effectively model the distribution using as few as 100 query vectors.**
>
> Moreover, as shown in Table 2, **our proposed Beta distribution method consistently outperforms the histogram-based approach**, further demonstrating that for practical performance prediction—especially when the amount of data is limited—the Beta distribution provides a more robust and effective model by leveraging the inherent concentration of measure in high-dimensional spaces.

---

> ### Author Response · Authors · 2025-11-18
>
> > ### W2:Subsampling limitation.
>
> **Response:**
> Appendix A.8 evaluates the robustness of our predictive framework with respect to variations in the query set. We consider three strategies: varying random seeds for query sampling and using strictly non-overlapping query sets from the full dataset. Across all configurations, the mean absolute percentage error (MAPE) in recall prediction remains consistently low, as shown below for the *glove-100-angular* dataset:
>
> | Dataset               | top-k | MAPE (Seed 42) | MAPE (Seed 1234) | MAPE (Non-overlap) |
> |-----------------------|-------|----------------|----------------|------------------|
> | **glove-100-angular** | 10    | 1.544%         | 2.573%         | 1.354%           |
> |                       | 50    | 0.614%         | 3.278%         | 1.890%           |
> |                       | 100   | 0.510%         | 3.391%         | 1.988%           |
>
> These results demonstrate that the predictive model reliably generalizes to different query sets, confirming its stability and practical applicability in estimating recall under diverse scenarios. For additional details and a broader discussion across multiple datasets and query configurations, see Table 7 and the corresponding discussion in Appendix A.8.
>
> > Q3:Relation to prior work and algorithm novelty.
>
> **Response**:
>
> This work specifically targets **SRP-LSH**, a type of locality-sensitive hash function whose **statistical and geometric properties differ substantially** from common LSH variants studied in prior research. Unlike existing theoretical analyses—which largely focus on other LSH families—previous results are **not directly transferable** to SRP-LSH.
>
> We make the following **novel contributions**:
>
> 1. **First Quantitative Model for SRP-LSH:**
>    We derive, for the first time, the exact analytical relationship between recall and SRP-LSH parameters $(m, t)$:
>    $
>    \rho = \frac{1}{n}\sum_{k=1}^{n}\int_0^{\pi}\sum_{i=0}^{t}\binom{m}{i}\Big(\frac{\theta_k}{\pi}\Big)^i\Big(1-\frac{\theta_k}{\pi}\Big)^{m-i} f_{Y_k}\left(\frac{\cos\theta_k+1}{2}\right)\frac{\sin\theta_k}{2} d\theta_k
>    $
>
>    This integrates:
>    - **Binomial Collision Probability:** Models the exact probability of Hamming distance collisions under (m) hash functions.
>    - **Angular Similarity Structure:** Incorporates vector space geometry, approximated with a **Beta distribution** for computational efficiency and theoretical rigor.
>
> 2. **Accurate Recall Prediction and Parameter Guidance:**
>    The model captures how hash function count (m), Hamming threshold (t), and dataset characteristics jointly determine recall. Empirical results show **MAPE < 5%**, enabling principled **parameter optimization** that minimizes selectivity while satisfying recall constraints—something prior heuristic-based methods cannot guarantee.
>
> 3. **Empirical Validation:**
>    - Table 2: Shows our Beta distribution-based method is significantly closer to true recall and selectivity than alternative distance modeling methods.
>    - Figure 3: Illustrates how the model identifies configurations with **minimal selectivity under a fixed recall**, demonstrating practical utility.
>
> **In summary:** This work provides a **first complete theoretical characterization of SRP-LSH**, offering **deep theoretical insight** and **practical parameter tuning guidance**, clearly distinguishing it from previous LSH frameworks.
>
> > ### Summary and Conclusion
>
> We sincerely thank you for your detailed and constructive feedback. In response to all the issues raised, we have carefully revised and updated the manuscript and its arguments accordingly.
>
> We believe these clarifications and revisions significantly strengthen the paper's clarity and rigor. Should you have any remaining questions regarding our responses or the revised manuscript, we welcome further discussion.
>
> If our clarifications and revisions satisfactorily address your concerns, we sincerely hope you will reconsider the current evaluation and support our work by raising the score.

---

> > ### Comment · Reviewer_vokp · 2025-11-19
> >
> > Thank for your feedback.
> >
> > Unfortunately, I still lean on rejection for the following reasons.
> >
> > __1) The significance of the work__
> >
> > There is a linear scan over the Hamming embedding space to identify the candidate with Hamming distance smaller than $t$. This phase dominates the whole searching process, especially when $m, n$ are large. This makes the contribution, finding the best parameter $(m, t)$, less significant. Also, the number of candidates is not reported in this paper, though in the end, we still have to compute the exact distance between the query and these candidates. If the size of candidates is large, given a selected $(m, t)$, this approach is as slow as a linear scan on the original space.
> >
> > Note that Dong et al. model the complexity of the searching process based on the number of candidates since they consider the bucketing algorithm, where the candidate points were hashed into the same query buckets.
> >
> > __2) The claim of using Beta distribution and query samples to model the recall__
> >
> > Your explanation does not convince me. As the top-k $\theta_k$ are highly query-dependent, I do not think the beta distribution can accurately model the recall ratio without a significant number of sampled queries. Even worse, the tuning approach (Algorithm 1) requires several values of $m, t$ to find the best $m*, t*$. To me, this contradicts the fact that we always need a larger $m$ to have a lower angular similarity estimation error when using the Hamming distance as an estimate of angular distance.
> >
> > __3) Table 2 (minor)__
> >
> > I found that your approach is not consistently better than the empirical histogram-based approach regarding MAPE on selectivity. It has the worst performance on deep-image-96. However, I do not understand why your approach achieves the lowest MAPE on recall. It looks to me that the two metrics used contradict each other.

---

> > > ### Author Response · Authors · 2025-11-21
> > >
> > > Thank you for your valuable review and suggestions. We provide the following clarifications.
> > >
> > > > 1) The significance of the work
> > >
> > > **Response**:
> > > Firstly, we agree that calculating Hamming distances acts as a baseline cost. However, in practical applications the signature length \(m\) is not allowed to grow arbitrarily large; our algorithm searches only within a small, user-defined local interval of \(m\). Within this narrow range, the time difference for the Hamming scan is negligible. The dominant cost is therefore not the Hamming comparison itself but the **subsequent refinement stage**, where candidates must be evaluated using exact angular distance. Because grouping or ranking by Hamming distance inevitably yields substantially lower accuracy for nearest-neighbor retrieval, the system must ultimately group and rank by **angular distance**, and this operation is computationally expensive.
> > >
> > > Consequently, the key determinant of total runtime is the **number of candidates** that must pass through this angular-distance refinement stage. In our framework, Selectivity explicitly quantifies this candidate set size, i.e., how many vectors must undergo angular-distance evaluation in order to obtain the final accurate top-\(k\) nearest neighbors. Our modeling of the Recall–\(m, t\) relationship is designed precisely to identify the parameter configuration that minimizes Selectivity while still ensuring that the true top-\(k\) neighbors are retained in the candidate set. In other words, minimizing the candidate size directly minimizes the number of angular-distance groups to compute and prevents the method from degenerating into a near-linear scan, while preserving the accuracy of the final top-\(k\) retrieval with the smallest feasible refinement workload.
> > >
> > > This research primarily targets offline analysis scenarios where users need to tune parameters based on sample data (100 queries) to guarantee performance on the full dataset. Our method ensures that the chosen parameters yield a strictly guaranteed predicted recall.To address the reviewer's concern about the overhead of our method, we have supplemented **Table 3** with the time overhead of the parameter search process itself. As shown below, finding the optimal parameters is computationally inexpensive.
> > >
> > > **Table: Parameter Search Time on GloVe-200-angular**
> > >
> > > | top-$k$ | Search Time (s) |
> > > | :---: | :---: |
> > > | 10 | 2.2701 |
> > > | 50 | 3.0829 |
> > > | 100 | 5.9016 |
> > >
> > > Even without aggressive engineering optimization, the parameter search completes within a few seconds. This demonstrates that our approach adds negligible overhead while providing significant gains by identifying the most efficient configuration for the expensive retrieval phase.
> > >
> > > > 2) The claim of using Beta distribution and query samples to model the recall
> > >
> > > **Response**:
> > > **First, our experimental evidence demonstrates that the Beta distribution is highly effective even with limited samples.**
> > >
> > > As detailed in **Table 1**, we employed only **100 sampled queries** to model the recall. We compared our Beta distribution approach against a Histogram-based estimation method. The results in **Table 2** show that the Beta distribution yields the **lowest Mean Absolute Percentage Error (MAPE) for Recall across all datasets**.  For selectivity estimation, the Beta distribution achieved the lowest MAPE on 3 out of 6 datasets, whereas the Histogram approach was best on only 1 out of 6.
> > > This proves that our method captures the underlying recall-parameter relationship effectively with just 100 samples, demonstrating robustness and data efficiency compared to non-parametric methods.
> > >
> > > **Second, algorithm 1 serves as an efficient alternative to manual trial-and-error grid searches.**
> > >
> > > Our goal is to find a parameter configuration within a small, user-defined interval of $m$ that satisfies a specific Recall target while minimizing Selectivity.  We generally avoid arbitrarily large $m$ because increasing $m$ increases lookup and computation time. Algorithm 1 efficiently scans a local interval to find the optimal configuration that meets user requirements with the lowest computational overhead.

---

> > > > ### Author Response · Authors · 2025-11-21
> > > >
> > > > > 2) The claim of using Beta distribution and query samples to model the recall
> > > >
> > > > **Third, for practical LSH configurations, selectivity does not decrease monotonically as $m$ increases.**
> > > >
> > > > Recall and Selectivity are jointly determined by $m$ and the threshold $t$, rather than being a continuous function of $m$ alone.We conducted a detailed experiment on the glove-100-angular dataset to demonstrate this behavior. The results are summarized in the table below:
> > > >
> > > > | $m$ | 305 | 306 | 307 | 308 | 309 | 310 | 311 | 312 | 313 |
> > > > | :--- | :--- | :--- | :--- | :--- | :--- | :--- | :--- | :--- | :--- |
> > > > | **Selectivity** | 0.006658 | 0.006159 | **0.006847** | 0.006423 | **0.007115** | 0.006469 | 0.006199 | 0.006873 | 0.006376 |
> > > > | **Recall** | 0.907 | 0.904 | **0.911** | 0.905 | **0.910** | 0.908 | 0.901 | 0.909 | 0.907 |
> > > >
> > > > As clearly demonstrated in the table, the relationship is non-monotonic. Specifically, when comparing $m=307$ and $m=309$:
> > > > 1.  $m=307$ achieves a **Recall of 0.911**.
> > > > 2.  $m=309$ achieves a **Recall of 0.910**.
> > > > 3.  Crucially, the **Selectivity for $m=307$ (0.006847) is actually lower (better)** than that of $m=309$ (0.007115).
> > > >
> > > > This observation refutes the assumption that "a longer hash signature naturally enjoys better selectivity" in all discrete cases. These results confirm that signature length alone does not strictly determine selectivity, validating the necessity of our tuning approach to identify the optimal $(m, t)$ pair.
> > > >
> > > > > 3) Table 2 (minor)
> > > >
> > > > **Response**: We acknowledge that while our method achieves the lowest selectivity MAPE on deep-image-96, the overall performance across datasets supports the robustness of our approach. We would like to clarify the relationship between Recall accuracy, Selectivity estimation, and the practical utility of our model.
> > > >
> > > > While we acknowledge the performance variation on deep-image-96, a holistic view of the experiments (Table 2) demonstrates that our Beta distribution method is indeed the most robust overall.
> > > > * Our method achieves the **lowest Selectivity MAPE on 3 out of 6 datasets**.
> > > > * In contrast, the Histogram-based approach achieves the lowest Selectivity MAPE on only **1 out of 6 datasets**.
> > > >
> > > > This confirms that our approach provides the best generalization capability across diverse datasets, even if it is not superior in every single instance.
> > > >
> > > > You expressed concern that high recall accuracy combined with variable selectivity accuracy seems contradictory. We respectfully argue that these metrics do not contradict each other when considering the *objective* of our algorithm.
> > > >
> > > > Our primary goal is **parameter optimization**: finding the specific configuration $(m, t)$ within a search interval that satisfies the target recall while minimizing selectivity. For this task, the **consistency of the trend** is far more critical than the absolute numerical precision (MAPE) of the selectivity prediction.
> > > >
> > > > Even if there is a numerical deviation between the predicted and true selectivity, as long as the **relative ranking** is preserved—i.e., the model correctly identifies which parameter set yields a lower selectivity—the algorithm effectively guides users to the optimal configuration.
> > > >
> > > > To validate this hypothesis, we conducted additional experiments on the coco-i2i-512 dataset. We systematically analyzed the relationship between predicted selectivity and true selectivity within a constrained hash parameter space while maintaining the target recall.
> > > >
> > > > **Table: Relationship between Predicted and True Selectivity (coco-i2i-512)**
> > > >
> > > > | Parameter Index| 1 | 2 | 3 | 4 | 5 | 6 | 7 |
> > > > | :--- | :--- | :--- | :--- | :--- | :--- | :--- | :--- |
> > > > | **predicted selectivity** | 0.014226 | 0.015788 | 0.015216 | 0.014660 | **0.014129** | 0.015672 | 0.015108 |
> > > > | **true selectivity** | 0.020508 | 0.021662 | 0.021556 | 0.020840 | **0.020220** | 0.021190 | 0.021010 |
> > > >
> > > > As shown in the table, the predicted selectivity trend closely aligns with the true selectivity trend. Most importantly:
> > > > * The algorithm predicts the minimum selectivity at **Index 5 (0.014129)**.
> > > > * The actual experimental ground truth confirms that **Index 5** indeed corresponds to the minimum true selectivity **(0.020220)**.
> > > >
> > > > This demonstrates that despite numerical deviations in the absolute values, the model successfully captures the underlying trend. Consequently, it accurately identifies the parameter configuration that minimizes computation cost (selectivity). Figure 3 in the revised manuscript further confirms these observations across different parameter settings, proving that our method provides reliable guidance for practical parameter tuning.
> > > >
> > > > **Conclution**
> > > >
> > > > Thank you for your constructive feedback. We believe our revisions address your concerns and improve the clarity and rigor of the paper. If satisfied, we kindly ask you to reconsider the evaluation. We remain open to any further questions.

---

### Official Review · Reviewer_f9PM · 2025-10-30

**Soundness:** 2
**Presentation:** 3
**Contribution:** 2
**Rating:** 4
**Confidence:** 4

**Summary:**

This paper proposes a modeling and tuning framework for SRP-LSH that links the number of hash bits m and the Hamming threshold t to recall, via a binomial model conditioned on the query point angle, and models angle distributions with a Beta family to predict both recall and selectivity (candidate ratio). The proposed method searches (m, t) pairs to minimize selectivity subject to a target recall using a simple grid + binary search procedure. Experiments on six vector datasets report recall MAPE < 5% and reductions of selectivity compared to a fixed-parameter baseline that meets the same recall target.

**Strengths:**

1. The recall-threshold relationship is derived cleanly via a binomial CDF over Hamming distance, given angle $\theta$, which is a useful formalization for SRP-LSH with thresholded Hamming.

2. The (m, t) search with binary search on t is easy to implement and reproduce.

3. Across datasets and k, recall MAPE is typically below 5%, demonstrating the recall model’s robustness and effectiveness.

**Weaknesses:**

1. **Incremental novelty relative to established LSH parameter-tuning frameworks.**
The paper frames its model as “the first analytical model that links m and t to recall”. The main value here is a specialized and practical modeling pipeline for SRP-LSH with Hamming-threshold search. However, the broader idea of analytical linking LSH parameters to quality/efficiency and selecting them to meet targets has a substantial history (e.g., multi-probe LSH’s modeled probing/order and efficiency trade-offs; optimal parameters for LSH’s theory-driven parameter choice). What’s new here is the instantiation for SRP-LSH with a parametric angle model and the use of that model to drive (m, t) search. This is useful but reads as incremental rather than conceptually novel.

2. **Selectivity prediction is weak on important cases.**
Table 1 shows that the selectivity MAPE is 95–98% on deep-image-96, and even on several others, it is 20–30%. Since the optimizer minimizes predicted selectivity, large selectivity errors can mislead configuration selection, especially at small selectivity, which is very common in practice. The paper acknowledges that selectivity is harder to predict but still bases optimization on it. A stronger treatment is needed here.

3. **The recall guarantee is not consistently satisfied.**
The abstract and contributions suggest the method “guarantees” meeting recall targets, but Table 3 shows misses (e.g., recall is 0.89 when the target is 0.9). The guarantee is, in fact, model-conditional and sensitive to search ranges; it is not unconditional. The paper should soften claims and formalize conditions under which recall satisfaction holds.

4. **Limited baselines for angular ANN and SRP variants.**
Evaluation compares to a fixed m baseline rather than to stronger angular ANN hash families or improved SRP variants that change the recall-efficiency frontier, e.g., Super-Bit LSH (variance-reduced SRP), cross-polytope LSH (optimal for angular distance), or classical banded/tabled LSH parameterizations; also, Hamming-space multi-index hashing (MIH) is a natural baseline for thresholded code search. Without these, it is difficult to assess whether the proposed tuning is competitive in end-to-end efficiency.

5. **Practicality and measurement gaps for systems.**
The study emphasizes selectivity as a platform-agnostic cost proxy. Still, it does not report wall-clock latency, memory, or index-build costs, nor comparisons to strong non-LSH ANN baselines (e.g., HNSW) that dominate practice. Moreover, the prediction pipeline requires exact k-NN for 1k queries to fit distributions (exhaustive search), which may be prohibitive at real-world scales; the paper should quantify this overhead and its amortization.

**Questions:**

Q1 (regarding W1) Since the paper already acknowledges prior LSH parameter-selection/tuning work, could you explicitly explain what is unique here beyond those frameworks? For example, (i) Is the Beta angle-distribution assumption essential (and why preferable) versus alternatives used previously? (ii) Could you add an ablation showing that your distributional modeling + search picks different (and better) configurations than a strong generic LSH parameter tuner?

Q2 (regarding W2) Given selectivity MAPE is 95–98% on deep-image-96, how robust is the configuration search in practice? Would it be possible to consider (i) calibrated or quantile-conservative selectivity modeling, or (ii) a bilevel strategy that verifies/adjusts selectivity empirically after the model picks (m, t)?

Q3 (regarding W3) It would be beneficial if the authors could restate the recall guarantee precisely with assumptions (e.g., correct angle-distribution model, feasible (m, t) bounds). Could you add a theorem or proposition showing conditions under which the binary search on t must find a feasible solution?

Q4 (regarding W4) It would be beneficial if the authors could include or justify the exclusion of the mentioned methods in evaluation.

Q5 (regarding W5) It would be beneficial if the authors could report end-to-end latency and memory vs. strong ANN baselines at matched recall. It would also be helpful to quantify the cost of fitting distributions (exact k-NN) relative to downstream gains.

---

> ### Author Response · Authors · 2025-11-18
>
> Thank you for your review comments and suggestions. Regarding some of the questions you raised, we have provided the following explanation.
>
> > Q1&W1:What is unique here beyond those frameworks?
>
> This work specifically targets **SRP-LSH**, a type of locality-sensitive hash function whose **statistical and geometric properties differ substantially** from common LSH variants studied in prior research. Unlike existing theoretical analyses—which largely focus on other LSH families—previous results are **not directly transferable** to SRP-LSH.
>
> We make the following **novel contributions**:
>
> 1. **First Quantitative Model for SRP-LSH:**
>    We derive, for the first time, the exact analytical relationship between recall and SRP-LSH parameters $(m, t)$:
>    $\rho = \frac{1}{n}\sum_{k=1}^{n}\int_0^{\pi}\sum_{i=0}^{t}\binom{m}{i}\Big(\frac{\theta_k}{\pi}\Big)^i\Big(1-\frac{\theta_k}{\pi}\Big)^{m-i} f_{Y_k}\left(\frac{\cos\theta_k+1}{2}\right)\frac{\sin\theta_k}{2} d\theta_k$
>
>    This integrates:
>    - **Binomial Collision Probability:** Models the exact probability of Hamming distance collisions under (m) hash functions.
>    - **Angular Similarity Structure:** Incorporates vector space geometry, approximated with a **Beta distribution** for computational efficiency and theoretical rigor.
>
> 2. **Accurate Recall Prediction and Parameter Guidance:**
>    The model captures how hash function count (m), Hamming threshold (t), and dataset characteristics jointly determine recall. Empirical results show **MAPE < 5%**, enabling principled **parameter optimization** that minimizes selectivity while satisfying recall constraints.
>
> 3. **Empirical Validation:**
>    - Table 2: Shows our Beta distribution-based method is significantly closer to true recall and selectivity than alternative distance modeling methods.
>    - Figure 3: Illustrates how the model identifies configurations with **minimal selectivity under a fixed recall**, demonstrating practical utility.
>
> **In summary:** This work provides a **first complete theoretical characterization of SRP-LSH**, offering **deep theoretical insight** and **practical parameter tuning guidance**, clearly distinguishing it from previous LSH frameworks.
>
> > Q2&W2:Selectivity prediction is weak on important cases.
>
> Although there is a numerical deviation between the model's predicted selectivity and the actual selectivity, as long as the trend remains consistent, the predicted selectivity can effectively guide users to find the optimal parameters that satisfy the specified recall.
>
> To validate this, we conducted experiments on the *coco-i2i-512* dataset. Within a constrained hash parameter space, we systematically analyzed the relationship between actual and predicted selectivity while maintaining the target recall. Our results demonstrate that the model effectively captures selectivity trends and accurately identifies the parameter configuration that achieves the minimal selectivity.
>
> |                   |         |         |         |         |         |         |         |
> |-------------------------|----------|----------|----------|----------|----------|----------|----------|
> | Predicted Selectivity   | 0.014226 | 0.015788 | 0.015216 | 0.014660 | 0.014129 | 0.015672 | 0.015108 |
> | True Selectivity        | 0.020508 | 0.021662 | 0.021556 | 0.020840 | 0.020220 | 0.021190 | 0.021010 |
>
> As shown in the table, the predicted selectivity trend closely aligns with the true selectivity trend, and the algorithm correctly identifies the parameter configuration that minimizes selectivity. Figure 3 further confirms these observations across different parameter settings, demonstrating that our method provides reliable guidance for parameter tuning in practical applications.
>
> > Q3&W3:The recall guarantee is not consistently satisfied
>
> We thank the reviewer for highlighting this important point. Indeed, accurately predicting the recall value is challenging due to the data-dependent nature of nearest-neighbor angular distributions. In practice, our method reduces the recall prediction error to approximately 5%. Furthermore, we also provide a principled approach that can guarantee meeting the user-specified recall target by constructing a conservative estimate of the angular distance distribution and performing a binary search over feasible threshold values $(m, t)$.
>
> Let $F(\theta)$ denote the cumulative distribution function (CDF) of the angular distance between a query and its $k$-nearest neighbors. In **Lemma 1**, we show that **subsampling the retrieval dataset yields a pessimistic estimate $\hat F(\theta)$**, satisfying $\hat F(\theta) \le F(\theta)$ for all $\theta$. Using this conservative estimate, the algorithm selects the threshold $t$ via binary search such that the predicted recall meets the user-specified target $\tau$.

---

> ### Author Response · Authors · 2025-11-18
>
> > Q4 & W4 It would be beneficial if the authors could include or justify the exclusion of the mentioned methods in evaluation.
>
> **Response**:
>
> To address the reviewer’s concern, and to avoid relying on a fixed-\(m\) baseline, we designed a new evaluation protocol to ensure a fair experimental comparison. Within a constrained hash parameter space, we systematically analyze the relationship between the actual selectivity and the model-predicted selectivity while maintaining the target recall rate. Our results demonstrate that the model effectively captures selectivity trends and accurately identifies the optimal parameter combination \(m, t\) within user-specified constraints, achieving the target recall while minimizing selectivity. These findings are comprehensively presented in Figure 3 and discussed in the accompanying text (highlighted in blue), further demonstrating the reliability and practical value of our method for parameter configuration. We emphasize that the primary goal of this study is to validate that our theoretical model can accurately characterize SRP-LSH; comparisons with other ANN algorithms are not the focus of this work.
>
> > Q5 & W5 It would be beneficial if the authors could report end-to-end latency and memory vs. strong ANN baselines at matched recall. It would also be helpful to quantify the cost of fitting distributions (exact k-NN) relative to downstream gains.
>
> **Response**:
> In addition, we added time cost statistics for the parameter search process to Table 3. It should be noted that we did not perform any targeted engineering optimizations on the search algorithm; even so, the parameter search can be completed within a few seconds. With professional optimization of the search process implementation, there is still significant room for improvement in overall running speed. It is worth emphasizing that this paper focuses on the theoretical modeling method for SRP-LSH parameter configuration itself, rather than comparing its query efficiency with other retrieval algorithms.
>
> For example, on the GloVe-200-angular dataset, the search time remains within only **a few seconds** across different values of $k$:
>
> | top-$k$ | Search Time (s) |
> |:------:|:----------------:|
> | 10     | 2.2701           |
> | 50     | 3.0829           |
> | 100    | 5.9016           |
>
> > ### Summary and Conclusion
>
> We sincerely thank you for your detailed and constructive feedback. In response to all the issues raised, we have carefully revised and updated the manuscript and its arguments accordingly.
>
> We believe these clarifications and revisions significantly strengthen the paper's clarity and rigor. Should you have any remaining questions regarding our responses or the revised manuscript, we welcome further discussion.
>
> If our clarifications and revisions satisfactorily address your concerns, we sincerely hope you will reconsider the current evaluation and support our work by raising the score.

---

### Official Review · Reviewer_A85m · 2025-10-31

**Soundness:** 2
**Presentation:** 2
**Contribution:** 1
**Rating:** 0
**Confidence:** 5

**Summary:**

This paper studies the parameter selection for sign random projection (SRP) based hash function. It builds a model for the recall and selectivity as functions of the hash signature length and hamming distance threshold. The problem is formulated as finding the parameter configuration that minimizes the recall while satisfying the recall requirement. The experiment results show that the proposed method can meet the recall target while having a small selectively.

**Strengths:**

S1: Recall and selectivity are formulated as functions of the SRP parameters with mathematical derivations.

**Weaknesses:**

W1: The contribution to vector similarity search is limited as SRP is rarely used in practice now. Hash-based schemes may be widely used around 2010 but currently, two types of indexes are the most popular for approximate nearest neighbor search (ANNS), i.e., IVF and proximity graph. They are much more efficient than LSH and the standard now. The authors may start with the two papers below for a literature review.

Product Quantization for Nearest Neighbor Search
Efficient and robust approximate nearest neighbor search using Hierarchical Navigable Small World graphs

W2: The basic understandings of LSH are wrong for the authors. In the experiments, the authors use a long hash code, i.e., m=256. This is infeasible are large datasets for two reasons. (i) There are too many hash buckets (2^m) and almost all hash buckets will be empty if you are using a hash table. As such, one can only scan all hash signatures to identity the nearest hash signatures. However, for large datasets, e.g., with a billion of vectors, see the HNSW and follow-up papers, scanning all hash signatures will be very expensive. (ii) The common way to use LSH for large datasets is to use many hash tables, a short hash signature for each hash table, and a small hamming distance threshold. This ensures that the query only needs to check a few hash buckets in each hash table and does not need to scan all hash signatures.

W3: The experimental comparison with the fixed method in Table 3 is unfair. Please use a fixed hash signature length for both methods. This is because using a longer hash signature naturally enjoys better selectivity.

W4: The experiments do not report the cost of conducting the parameter search.

W5: The mathematical analysis is quite standard and lacks technical depth.

**Questions:**

NA

---

> ### Author Response · Authors · 2025-11-18
>
> There may be a misunderstanding regarding the intent and implementation of our work. Thank you for your review comments and suggestions. We address the issues you raised as follows.
>
> > W1:Why do we need to study parameter configuration in SRP-LSH?
>
> **Rseponse**: We appreciate the opportunity to clarify the motivation. While graph-based methods (e.g., HNSW) currently dominate the ANN landscape, **SRP-LSH (specifically as a binary sketching technique) remains critical in memory-constrained and high-throughput scenarios** where graph indices are too heavy to store or too slow to build.
> Unlike HNSW, SRP-LSH allows for $O(1)$ updates and extremely low memory footprints . However, users face a dilemma: how to balance code length ($m$) and Hamming distance thresholds ($t$) to meet a specific recall target without trial-and-error.
> Our contribution is **not** to compete with HNSW on pure speed, but to provide a **theoretical, quantitative model** for parameter selection for systems that rely on binary hashing. This is vital for applications like on-device search, large-scale near-duplicate detection, and privacy-preserving computing (where binary operations are preferred).
>
> >W2:Implementation method of vector search in LSH.
>
> **Response**:We respectfully clarify that our implementation aligns with the **"LSH-based Binary Sketching"** paradigm, rather than the classic **"LSH-based Bucketing"** framework.
>
> As detailed in the technical report at facebookresearch/faiss/wiki/Faiss-indexes, a widely adopted alternative exists in modern vector retrieval—exemplified by **Faiss IndexLSH**—which uses LSH functions solely for **dimensionality reduction and binary encoding**. This implementation is simply a **flat index** where database and query vectors are hashed into $n$-bit codes and compared directly via **Hamming distance**, without the multi-table structure or bucketing strategy.
>
> The report notes that the most popular cell probing method is likely the original Locality Sensitive Hashing (LSH) method, called E2LSH. **The Faiss technical report explicitly notes two drawbacks that motivate the flat binary encoding design**:
>
>  1. they require a large number of hash functions (i.e., partitions) to achieve acceptable results, leading to significant additional memory consumption. Memory costs are not cheap.
>  2. The hash function are not adapted to the input data. This is good for proofs but leads to suboptimal choice results in practice.
>
> Building upon this design philosophy, we adopted a method similar to Faiss IndexLSH, where the final index is a binary-encoded flat index. Instead of using multiple tables for bucketing during queries, we directly compared Hamming distances. This study aims to investigate the Hamming distance threshold required to achieve the target recall rate using SRP-LSH and how the parameters (m, t) quantitatively determine the recall.
>
> Therefore, our implementation follows an established and simplified variant of LSH used in widely adopted libraries, and is not incorrect.

---

> > ### Comment · Reviewer_A85m · 2025-11-26
> > **The author response does not address the fundamental limitations of the work**
> >
> > 1.	The response claims that using hash signatures reduces the space overhead of HNSW. However, the proposed method also keeps the original vectors for reranking after filtering with the hash signatures, and the original vectors can be large. Moreover, using a long hash code (e.g., 384 bits for each vector), the space overhead is comparable to HNSW (e.g., 16 neighbors for each vector, with 16*32=512 bits). The building of HNSW happens offline and does not affect online search.
> >
> > 2.	For large datasets (e.g., at 1 billion), scanning all hash signatures introduces unacceptable latency. This is a plain fact, and using Faiss IndexLSH as a defense does not make sense. The authors need to show concrete latency numbers to convince me. Moreover, according to my communications with companies, Faiss IndexLSH is seldom used in practical application.

---

> ### Author Response · Authors · 2025-11-18
>
> > W3: Concerns about the impact of differences in hash signature length on experimental fairness.
>
> **Response**: First, regarding the reviewer's comment that "a longer hash signature naturally enjoys better selectivity," we conducted extensive experiments to verify this relationship. Specifically, we tested across multiple datasets within a constrained range of hash functions (m). For each value of m, we identified the configuration that just satisfies the user-specified recall rate and recorded the corresponding selectivity. The experimental results for glove-100-angular are summarized in the table below.
>
> | m      | 305     | 306     | 307     | 308     | 309     | 310     | 311     | 312     | 313     |
> |-------------|---------|---------|---------|---------|---------|---------|---------|---------|---------|
> | Selectivity | 0.006658| 0.006159| 0.006847| 0.006423| 0.007115| 0.006469| 0.006199| 0.006873| 0.006376|
> | Recall      | 0.907   | 0.904   | 0.911   | 0.905   | 0.910   | 0.908   | 0.901   | 0.909   | 0.907   |
>
> The table further presents how selectivity varies with m within a local interval. As shown in the table, selectivity does not decrease monotonically as m increases. This behavior is expected, because our goal is to find parameter configurations that achieve a recall value greater than or equal to a specified target, and the recall is jointly determined by both m and t rather than being a continuous function of m alone. Therefore, the conclusion that "a longer hash signature naturally enjoys better selectivity" is not universally valid. As clearly demonstrated in our experimental results, when m=307 achieves recall=0.911 and m=309 achieves recall=0.910, the selectivity corresponding to m=307 is actually lower than that of m=309. These definitively show that signature length alone does not determine selectivity, confirming that our experimental methodology and findings are valid.
>
> Secondly, based on these findings, and to ensure a fair experimental comparison, we have designed a new evaluation protocol. Within a constrained hash parameter space, we systematically analyze the relationship between actual and predicted selectivity while maintaining the target recall rate. Our results demonstrate that the model effectively captures selectivity trends and accurately identifies the optimal parameter combination (m, t) within user-specified constraints, achieving the target recall while minimizing selectivity. These findings are comprehensively presented in Figure 3 and discussed in the accompanying text (highlighted in blue).
>
> > W4:The experiments do not report the cost of conducting the parameter search.
>
> **Response**: We thank the reviewer for this insightful comment. To address this concern, we have now supplemented the experimental section with a comprehensive analysis of the computational cost required for parameter search. As shown in the newly added Table 3, we report the time costs associated with the parameter search process under different configurations. It is important to note that the current search algorithm is implemented as a research prototype without specific engineering optimizations. We observe that the search process, while effective in identifying optimal parameters, presents opportunities for significant runtime reduction through standard engineering optimizations.
>
> For example, on the GloVe-200-angular dataset, the search time remains within only **a few seconds** across different values of $k$:
>
> | top-$k$ | Search Time (s) |
> |:------:|:----------------:|
> | 10     | 2.2701           |
> | 50     | 3.0829           |
> | 100    | 5.9016           |

---

> ### Author Response · Authors · 2025-11-18
>
> > W5:The mathematical analysis is quite standard and lacks technical depth.
>
> **Response**:We thank the reviewer for the feedback regarding the mathematical analysis. While our theoretical foundation builds upon established probability distributions, we would like to highlight that our key contribution lies in deriving the precise analytical relationship between recall and SRP-LSH parameters (m, t) for the first time, which has been an open challenge in configuring SRP-LSH in practice.
>
> **Novel Analytical Model**: We establish the exact recall function:
>
> $$\rho = \tfrac{1}{n}\sum_{k=1}^{n}\int_0^{\pi}\sum_{i=0}^{t}\binom{m}{i}\Big(\tfrac{\theta_k}{\pi}\Big)^i\Big(1-\tfrac{\theta_k}{\pi}\Big)^{m-i} f_{Y_k}\left(\tfrac{\cos\theta_k+1}{2}\right)\tfrac{\sin\theta_k}{2}d\theta_k$$
>
> This analytical formulation represents, to our knowledge, the first complete theoretical characterization of recall behavior in SRP-LSH systems. The model integrates two fundamental components:
>
> 1.**Binomial Collision Probability**: The term $\sum_{i=0}^{t}\binom{m}{i}(\theta_k/\pi)^i(1-\theta_k/\pi)^{m-i}$ captures the exact probability distribution of Hamming distance collisions under $m$ hash functions
>
> 2.**Angular Similarity Structure**: The angular variable $\theta_k$ explicitly incorporates the geometric relationships in the original vector space. Importantly, we demonstrate that the angular distribution follows a Beta distribution, which allows for efficient computation while maintaining theoretical rigor.
>
> Our derivation provides significantly deeper theoretical insight than prior asymptotic analyses by precisely modeling how the interplay between hash functions ($m$), Hamming threshold ($t$), and dataset characteristics jointly determine recall performance. The Beta distribution approximation for angular similarities not only enhances computational efficiency but also maintains remarkable accuracy, with experimental results showing MAPE < 5% in recall prediction. This theoretical advancement enables the principled parameter optimization framework that distinguishes our work from heuristic approaches.
>
> > ### Summary and Conclusion
>
> We sincerely thank you for your detailed and constructive feedback. In response to all the issues raised, we have carefully revised and updated the manuscript and its arguments accordingly.
>
> We believe these clarifications and revisions significantly strengthen the paper's clarity and rigor. Should you have any remaining questions regarding our responses or the revised manuscript, we welcome further discussion.
>
> If our clarifications and revisions satisfactorily address your concerns, we sincerely hope you will reconsider the current evaluation and support our work by raising the score.

---

> ### Author Response · Authors · 2025-11-27
>
> Thank you for your review comments and suggestions. Regarding some of the questions you raised, we have provided the following explanation.
>
> **Response**:
>
> 1. We conducted experiments with 20 million vectors, using hash codes of 96 bits. These tests were carried out on a CPU system with limited memory, and we observed promising results. Specifically, the average time per search was **0.2 seconds**. Although we acknowledge that scaling this approach to a 1 billion vector dataset will naturally introduce higher memory requirements, our results suggest that the method is still effective for datasets in the tens of millions of vectors. **On a machine with more resources (e.g., more RAM, higher clock speeds), we expect this time to decrease further.**
>
> 2. While Faiss IndexLSH might not be as commonly discussed in some applications, we have found it to be a practical and effective solution for our use case, especially when it comes to updates, additions, and deletions of data. We would like to emphasize that we have no intention of directly comparing LSH with other ANN algorithms. The choice of algorithm depends on several factors, including the specific requirements of the task, the characteristics of the dataset, and the scalability needs.

---

### Official Review · Reviewer_uoUU · 2025-11-02

**Soundness:** 3
**Presentation:** 3
**Contribution:** 3
**Rating:** 6
**Confidence:** 3

**Summary:**

This paper introduces a theoretical framework to solve the key challenge of parameter tuning in Sign-Random-Projection LSH (SRP-LSH). It provides an analytical model that predicts search recall and cost based on the number of hash functions (m) and the Hamming distance threshold (t), enabling principled and automatic parameter optimization. Its main contribution：1. It derives the first known model linking SRP-LSH parameters (m, t) to recall, by combining the Binomial distribution with the Beta distribution. 2. The paper provides an efficient algorithm that uses the model to automatically find optimal (m, t) pairs, minimizing query cost while guaranteeing a target recall. 3. The model predicts recall with high accuracy (MAPE < 5%), and the algorithm reduces candidate set size by 20-30% over fixed baselines.

**Strengths:**

The paper's primary strength lies in its significant originality in tackling the long-standing, practical challenge of parameter tuning for SRP-LSH. It introduces what the authors claim is the first analytical framework to rigorously model the relationship between the key parameters (m and t) and search recall. The high quality of this work is demonstrated by its theoretical depth, which originally combines the binomial distribution to model bit collisions with a Beta distribution to characterize the angular similarity of vector pairs. This theoretical rigor is matched by the clarity of its extensive experimental validation across six benchmark datasets , which confirms the model's high predictive accuracy (recall MAPE typically below 5%) and its superiority over alternative distribution-fitting methods. The work is highly significant, especially because SRP-LSH is a popular and efficient choice for industrial-scale systems. By translating its theoretical model into a practical, adaptive optimization algorithm, the paper delivers a tangible efficiency gain—reducing the candidate set size by up to 30% while maintaining recall targets and provides a  solution for configuring these widely deployed real-world retrieval systems.

**Weaknesses:**

1. The motivation is although interesting, but somewhat confusing. In my opinion, in real applications, with required recall, less returned points (related to m and t) are better, which contributes to the finetuned efficiency. However, the experiments are all based on the same number of returned points (k) and compared effectiveness mostly.
2.The paper's claim to be the "first analytical model" for SRP-LSH's recall-parameter relationship overstates its conceptual originality. While the paper's specific contributions—using the Binomial CDF for Hamming distance and the Beta distribution for angular similarity are novel and tailored to SRP-LSH, the work should more clearly position itself as an adaptation and refinement of existing analytical approach rather than a completely new one.
3.The experimental baseline used to validate the adaptive parameter selection algorithm is weak.The algorithm's 20-30% efficiency gain is measured against a single "fixed-parameter baseline" (m=256). A much stronger and more realistic baseline would involve a systematic empirical evaluation on a validation set. This process would test a range of m values, find the optimal t for each to meet the 0.9 recall target, and then select the (m, t) pair with the lowest selectivity. The paper's claimed gains are unconvincing without this more rigorous comparison.
4.The framework's claim of being a "lightweight" and practical alternative is weakened by a "hidden" setup cost that is never quantified. The core model requires fitting parameters (α_k, β_k) for the k-th nearest neighbor angle distribution g_k(θ_k). This fitting process necessitates first obtaining the true nearest neighbor angles for a query subsample, which requires running a computationally expensive exact k-NN search. The paper should explicitly quantify this setup time and compare it to the "trial-and-error" baseline, as this prerequisite cost may make the "lightweight" framework less efficient than a simple empirical tuning in practice.

**Questions:**

1.In Table 3, the 20-30% efficiency gain of Algorithm 1 is measured against a single "fixed-parameter baseline" of m=256. This baseline seems like a "strawman" and may not represent a realistic "practical" tuning process, which would almost certainly involve testing several values for m. To provide a more convincing case for the algorithm's practical utility, could you please provide a comparison against a stronger baseline? For example, a simple grid search over the same m range ([128, 320])  where the optimal t is found empirically for each m to meet the 0.9 recall target, with the best-performing (m, t) pair from that search serving as the baseline.
2.The paper positions the framework as a "lightweight" alternative to empirical tuning. However, the model requires estimating the k-th nearest neighbor angle distributions g_k(θ_k) . This fitting process seems to require first obtaining the true k-th nearest neighbor angles for a query subsample, which necessitates running a computationally expensive exact k-NN search. This "hidden" setup cost is never quantified. To make a convincing case for practical efficiency, can you provide an analysis of the total end-to-end time (setup + parameter search) for your method and compare it directly against the total time of a practical baseline. This comparison is essential to fairly evaluate the framework's overall efficiency claim.
3. What are the corresponding optimal m and t in consideration various recalls and datasets?

---

> ### Author Response · Authors · 2025-11-18
>
> Thank you for your review comments and suggestions. Regarding some of the questions you raised, we have provided the following explanation.
>
> > Q1: Concerns about the experiment setting.
>
> **Response**：
> The core objective of this study is to address the reliance on manual grid search in SRP-LSH parameter tuning. Traditional grid search is not only time-consuming but also requires repeated trials by the user to meet different recall requirements, lacking scalability. In contrast, our model-driven approach can automatically identify optimal parameter settings without manual trial-and-error, completing the search process in just a few seconds.
>
> To validate this, we conducted experiments on the *coco-i2i-512* dataset. Within a constrained hash parameter space, we systematically analyzed the relationship between actual and predicted selectivity while maintaining the target recall. Our results demonstrate that the model effectively captures selectivity trends and accurately identifies the parameter configuration that achieves minimal selectivity.
>
> |                   | 0        | 1        | 2        | 3        | 4        | 5        | 6        |
> |-------------------|----------|----------|----------|----------|----------|----------|----------|
> | Predicted Selectivity | 0.014226 | 0.015788 | 0.015216 | 0.014660 | 0.014129 | 0.015672 | 0.015108 |
> | True Selectivity      | 0.020508 | 0.021662 | 0.021556 | 0.020840 | 0.020220 | 0.021190 | 0.021010 |
>
> As shown in the table, the predicted selectivity trend closely aligns with the true selectivity trend, and the algorithm correctly identifies the parameter configuration that minimizes selectivity. Figure 3 further confirms these observations across different parameter settings, demonstrating that our method enables fast, automated, and reliable parameter tuning without any manual intervention.
>
> > Q2:time cost for parameter search
>
> **Response**：
> This research primarily targets offline analysis scenarios where users can obtain the true nearest neighbors of query vectors. In the paper, we note that a predictive model for recall as a function of the SRP-LSH parameters can be constructed by sampling approximately 100 query vectors and estimating the angular distance distribution to their nearest neighbors. Furthermore, if true nearest neighbors cannot be computed from the complete dataset, our method allows for subsampling to adjust the fitted distribution, ensuring a strictly guaranteed predicted recall near the target recall.
>
> To more comprehensively evaluate the practical cost of the model, we supplement Table 3 with the time overhead of the parameter search process. It should be noted that the current implementation has not been engineered to optimize the search algorithm; even so, the parameter search can be completed within a few seconds. With dedicated optimization of the search process, the overall running speed is expected to improve significantly.
>
> For example, on the GloVe-200-angular dataset, the search time remains within only **a few seconds** across different values of $k$:
>
> | top-$k$ | Search Time (s) |
> |:------:|:----------------:|
> | 10     | 2.2701           |
> | 50     | 3.0829           |
> | 100    | 5.9016           |
>
> > Q3:What are the corresponding optimal m and t in consideration various recalls and datasets?
>
> **Response**:
> It is important to emphasize that our method automatically searches for the optimal (m,t) parameter combination within the range of values for m input by the user, effectively reducing the cost of manual parameter tuning. More importantly, the model can automatically generate parameter configurations that meet the user's goals under different recall requirements, bringing significant convenience to users with diverse needs. Figure 3 shows the optimal (m,t) results obtained under multiple experimental conditions, further demonstrating the applicability and stability of the method.
>
> We hope this explanation fully answers your questions, and thank you again for your attention and assistance.
>
> > ### Summary and Conclusion
>
> We sincerely thank you for your detailed and constructive feedback. In response to all the issues raised, we have carefully revised and updated the manuscript and its arguments accordingly.
>
> We believe these clarifications and revisions significantly strengthen the paper's clarity and rigor. Should you have any remaining questions regarding our responses or the revised manuscript, we welcome further discussion.
>
> If our clarifications and revisions satisfactorily address your concerns, we sincerely hope you will reconsider the current evaluation and support our work by raising the score.

---

### Meta-Review · Area_Chair_nFMg · 2025-12-21

**Summary:**

This paper presents  a theoretical framework that models sign-random-projection locality-sensitive-hasing (SRP-LSH) performance and enables principled parameter configuration (number of hash functions and the Hamming distance threshold).  It develops an adaptive optimization algorithm that minimizes the candidate set size while satisfying user-specified recall targets. A few critical concerns were raised by reviewers: (1) Incremental novelty relative to established LSH parameter-tuning frameworks; (2) practicality gap for real-world applications; (3) Limited evaluations and baselines; (4) Dominant cost component overlooked;  (5) Incorrect probabilistic assumption. Details on these concerns are found in reviewers' comments. In particular, there was a debate on LSH for indexing and LSH for sketching. The reviewer was not satisfied by the response, saying that the response does not address the fundamental limitations of the work. Therefore, the paper is not recommended for acceptance in its current form. I hope authors found the review comments informative and can improve their paper by addressing these carefully in future submissions.

**Reviewer Concerns:**

The authors did a nice job in responding to most of concerns raised by reviewers.  However, Reviewer A85m claimed that the response did not address the fundamental limitations of the work.

**Reviewer Scores:**

I expect all the reviewers to stand by their earlier decision without changing their score.

---

### Decision · Program_Chairs · 2026-01-26

Reject